# Metabolic Dysfunction and Dietary Interventions in Migraine Management: The Role of Insulin Resistance and Neuroinflammation—A Narrative and Scoping Review

**DOI:** 10.3390/brainsci15050474

**Published:** 2025-04-29

**Authors:** Cinzia Cavestro

**Affiliations:** Headache Centre, Department of Neurology, ASL CN2, Community Health Center—Former San Lazzaro Hospital, 12051 Alba, Italy; ccavestro@aslcn2.it

**Keywords:** migraine, inflammation, neurogenic inflammation, insulin, diet, dietary

## Abstract

Introduction: Migraine is a prevalent neurological disorder characterized by recurrent headaches with autonomic and neurological symptoms, significantly impacting quality of life globally. Its pathogenesis involves genetic, neurological, inflammatory, and metabolic factors, with insulin resistance and metabolic dysfunction increasingly recognized as important contributors. Historically, it has been known that certain foods can trigger migraine attacks, which led for many years to the recommendation of elimination diets—now understood to primarily target histamine-rich foods. Over the past two decades, attention has shifted toward underlying metabolic disturbances, leading to the development of dietary approaches specifically aimed at addressing these dysfunctions. Methods: A scoping literature review was conducted using PubMed and Embase to evaluate the relationships among migraine, insulin-related mechanisms, neurogenic inflammation, and dietary interventions. Initial searches focused on “MIGRAINE AND (neurogenic inflammation)” (2019–15 April 2025), followed by expanded searches from 1950 onward using terms such as “MIGRAINE AND (insulin, insulin resistance, hyperinsulinism)”, and “MIGRAINE AND (diet, dietary, nutrition, nutritional)”. A specific search also targeted “(INSULIN OR insulin resistance OR hyperinsulinism) AND (neurogenic inflammation)”. Abstracts were screened, full texts were retrieved, and duplicates or irrelevant publications were excluded. No filters were applied by article type or language. Systematic reviews and meta-analyses were prioritized when available. Results: Migraine pathogenesis involves trigeminovascular system activation, neurogenic inflammation mediated by CGRP and PACAP, immune dysregulation, mast cell activation, and cortical spreading depression (CSD). Emerging evidence highlights significant associations between migraine, insulin resistance, and hyperinsulinism. Hyperinsulinism contributes to migraine through TRPV1 sensitization, increased CGRP release, oxidative stress, mitochondrial dysfunction, and systemic inflammation. Metabolic dysfunction, including obesity and insulin resistance, exacerbates migraine severity and frequency. Dietary interventions, particularly anti-inflammatory, Mediterranean, and ketogenic diets, show promise in reducing migraine frequency and severity through mechanisms involving reduced inflammation, oxidative stress, improved mitochondrial function, and glucose metabolism stabilization. Conclusions: The interplay between insulin resistance, metabolic dysfunction, and neuroinflammation is crucial in migraine pathophysiology. Targeted dietary interventions, including ketogenic and Mediterranean diets, demonstrate significant potential in managing migraines, emphasizing the need for personalized nutritional strategies to optimize therapeutic outcomes.

## 1. Introduction

Migraine is a prevalent and disabling neurological disorder characterized by recurrent episodes of moderate to severe headaches, often accompanied by a range of autonomic and neurological symptoms. Affecting approximately 1 billion people globally, migraines are among the leading causes of disability, particularly in individuals aged 15–49 years [1]. The condition is more common in women than in men, likely influenced by hormonal fluctuations [2]. Based on large population-based studies, migraine prevalence is 12% in the general population, 18% in women, and 6% in men when stratified by gender, with 1% suffering from chronic migraine [3]. Migraine is classified as a primary headache disorder, with diagnostic criteria outlined in the International Classification of Headache Disorders [4]. It is fundamentally divided into two forms: migraine without aura and migraine with aura. Migraine without aura is described as a recurrent headache disorder manifesting in attacks lasting 4–72 h. Typical characteristics of the headache are unilateral location, pulsating quality, moderate or severe intensity, aggravation by routine physical activity, and association with nausea and/or vomiting and/or photophobia and phonophobia. Migraine with aura is described as recurrent attacks (at least two), lasting 5–60 min, of unilateral fully reversible visual, sensory, or other central nervous system symptoms that usually develop gradually and are usually followed by headache and associated migraine symptoms [4].

The pathogenesis of migraines involves a complex interaction of genetic, neurological, and environmental factors [5].

Dysregulated neuronal activity triggers activation of the trigeminovascular system, leading to neurogenic inflammation, cerebral vasodilation, and pain [6]. Activation of the trigeminovascular system plays a central role, with the release of pro-inflammatory neuropeptides such as calcitonin gene-related peptide (CGRP) contributing to the characteristic throbbing pain and associated symptoms like photophobia, phonophobia, nausea, and aura [7].

Over 180 genetic variations have been identified, many of which are linked to neuronal or vascular dysfunctions that contribute to migraine susceptibility [8].

Key risk factors include alcohol dependence, smoking, depression, anxiety, insomnia, sleep apnea, obesity, and glucocorticoid use [9].

Factors such as alcohol dependence [10,11], smoking [12], depression [13,14], anxiety [15], insomnia or sleep apnea [16], obesity [17], and glucocorticoid use [18,19], have been implicated in migraine pathogenesis.

These factors are thought to influence migraine pathogenesis by disrupting neurotransmitter systems, hormones, and immune regulation [20].

Inflammation and immune activation play a central role in migraines. Elevated levels of inflammatory cytokines, such as IL-6, TNF-α, and CGRP, are implicated in pain transmission and vascular changes during migraine episodes. Immune cells, including T cells and macrophages, further sustain an inflammatory environment, exacerbating symptoms [21,22,23].

Epidemiological studies show reciprocal associations between migraine and risk factors like depression, anxiety, and obesity, suggesting shared genetic and physiological pathways involving inflammation and hormonal changes [16,24].

Migraine frequently coexists with a range of comorbid conditions, including psychiatric disorders (such as depression and anxiety), cardiovascular diseases, metabolic syndromes, and other neurological disorders [3,25].

The presence of these comorbidities often complicates the clinical picture and can influence both the frequency and severity of migraine attacks [26]. For instance, individuals with migraine are at an increased risk of developing depression, while those with depression are more likely to experience migraines. This bidirectional relationship suggests overlapping pathophysiological mechanisms, including alterations in serotonin pathways and hypothalamic–pituitary–adrenal axis dysfunction [26,27].

Addressing comorbidities is an essential aspect of effective migraine management. A comprehensive assessment should include routine screening for common comorbid conditions, as their identification can guide treatment strategies [https://americanmigrainefoundation.org/resource-library/migraine-comorbidities (accessed on 8 February 2025)]. For example, tricyclic antidepressants, such as amitriptyline, are effective for patients with coexisting depression and migraine, providing dual therapeutic benefits [28]. Similarly, lifestyle interventions, including stress management, regular physical activity, and dietary modifications, can alleviate symptoms of both migraine and its comorbidities [29].

An integrated approach to the management of migraine and its comorbidities often necessitates a multidisciplinary framework, involving neurologists, psychiatrists, cardiologists, and other healthcare providers. This collaborative strategy ensures that all dimensions of a patient’s health are addressed, leading to more effective and personalized care. By treating both migraine and its comorbidities, healthcare providers can reduce overall disease burden and enhance patient quality of life [30].

Metabolic comorbidities, including obesity, insulin resistance, type 2 diabetes mellitus, and dyslipidemia, are increasingly recognized as important factors in the pathogenesis and progression of migraine [31,32]. These conditions share overlapping mechanisms with migraine, such as systemic inflammation, oxidative stress, and alterations in energy metabolism. For example, adipose tissue-derived pro-inflammatory cytokines and neuropeptides like leptin and adiponectin may influence migraine susceptibility and attack frequency [33,34].

Obesity, in particular, has been associated with an increased risk of both chronic and episodic migraines. Studies suggest a dose–response relationship, where a higher body mass index (BMI) correlates with greater migraine frequency and severity. Central obesity, a marker of metabolic syndrome, further exacerbates this relationship through enhanced systemic inflammation and vascular dysregulation [35].

Dietary interventions aimed at addressing metabolic dysfunction have emerged as promising strategies for managing migraines [32]. For instance, the ketogenic diet, characterized by a low carbohydrate intake and high fat consumption, has shown potential in reducing migraine frequency. This effect is hypothesized to stem from the stabilization of neuronal energy metabolism, anti-inflammatory effects, and the modulation of glutamate and gamma-aminobutyric acid (GABA) pathways [36].

Furthermore, weight loss achieved through calorie restriction or bariatric surgery has been associated with significant reductions in migraine frequency and severity [37,38]. Beyond calorie control, dietary patterns that emphasize anti-inflammatory foods, such as the Mediterranean diet, can also benefit migraineurs with metabolic comorbidities by reducing systemic inflammation and improving vascular health [37,38].

Identifying and addressing metabolic comorbidities in migraine patients is crucial for optimizing outcomes. Incorporating dietary strategies tailored to individual metabolic profiles not only aids in migraine management but also mitigates the risk of long-term complications associated with metabolic disorders.

### The Scope of This Paper

The aim of this article is to provide an updated overview of the relationship between insulin and migraine, with both of them linked to inflammation, as well as to evaluate the potential therapeutic mechanisms of dietary interventions for migraine management. The article will explore neurogenic inflammation in migraine, how insulin is involved in neurogenic inflammatory processes, what is known about insulin alterations in migraineurs, and finally, the potential impact of diet in modulating these processes.

Other aspects of diets or using supplements to treat migraine are not among the main aims of this paper.

## 2. Methods

Neurogenic inflammation is a very interesting phenomenon I dealt with a few years ago [23]. An update has been made for ‘MIGRAINE AND (neurogenic inflammation)’, using PubMed database (all references since 2019 until 15–31 April 2025). After identifying key thematic areas, a broader and more targeted search was performed using both PubMed and Embase databases, covering all references from 1950 to 15 April 2025. Figure 1 illustrates the process of the research, the distribution of articles across these two databases, and a flowchart of paper selection. The literature review was based on the following search terms: MIGRAINE AND (insulin or insulin resistance or hyperinsulinism); MIGRAINE AND (diet, dietary, nutrition, nutritional).

A specific further search was performed for (INSULIN or INSULIN RESISTANCE or HYPERINSULINISM) AND (neurogenic inflammation) to look for research on the possible role of insulin in neurogenic inflammation and migraine.

These search terms were used to identify studies relevant to the interaction between migraine, insulin-related mechanisms, neurogenic inflammation, and dietary interventions. Abstracts were screened, and all potentially relevant full-text articles were retrieved. After the exclusion of duplicates and irrelevant publications, the remaining articles were used to develop the present scoping review. No filters were applied with regard to article type or language. If any systematic reviews with meta-analyses were found, they were taken as the primary reference articles.

Exclusion of duplicates refers to the process of identifying and removing multiple records of the same article that may appear more than once across different databases (e.g., PubMed, Embase) or within the same database due to variations in citation formatting or indexing. A study was considered relevant if it provided clinical, experimental, or epidemiological data related to one or more of the above domains; explored the mechanisms linking metabolic dysfunction or inflammation to migraine; and evaluated the effectiveness of diet-based therapeutic strategies in migraine management. Conversely, studies were considered non-relevant if they did not address migraine in the context of the selected search themes; were theoretical or editorial in nature without empirical data; and focused solely on unrelated pathologies, or were case reports not generalizable to the scope of this review.

## 3. Results

After the screening process described in the Methods Section, a total of 4918 articles were considered. The study design and the PRISMA flowchart for the scoping review are presented in Figure 1.

Given the complexity of the topic, the results are organized into the following sections: an updated overview of migraine and neurogenic inflammation; metabolic alterations identified in migraineurs; the association between insulin dysregulation and neurogenic inflammation; and, finally, the role of dietary interventions in the management of migraine.

### 3.1. Migraine and Neurogenic Inflammation

The pathophysiology of migraine primarily involves the activation of trigeminal sensory neurons, initiated in deep brain structures [39,40], and engaging both central neurons [41] and peripheral nociceptors located in the dura mater [42] and cerebral arteries [43]. This complex network highlights the role of peripheral sensitization in migraine pathogenesis, with abnormal activation confined to susceptible individuals [44].

Pain originates from the sensory innervation of cranial meningeal arteries and venous sinuses, primarily supplied by the trigeminal ganglion and cervical roots. Evidence suggests that the activation of meningeal nociceptors is a fundamental mechanism of migraine attacks, resembling headaches arising from intracranial pathologies [44].

Historically, the vascular theory of migraine, which proposed an aberrant response of the cerebral vasculature, was the dominant explanation [44,45,46]. However, this view evolved after Dr. Moskowitz’s 1984 study [47], which demonstrated the involvement of the trigeminovascular system and shifted the focus toward neurochemical mechanisms [48].

The theory of neurogenic inflammation, introduced by Moskowitz in 1979 [49], emphasized the roles of neuropeptides such as substance P (SP), neurokinin A (NKA), and calcitonin gene-related peptide (CGRP) in promoting vasodilation, edema, and plasma extravasation—key contributors to migraine pathogenesis. Contemporary research now emphasizes the interplay between vascular, neurochemical, and inflammatory processes in migraine development and its therapeutic implications. Two major reviews have been published on this topic: the first by Dr. Moskowitz in 1993 [50], and a second by Akerman et al., two decades later [51], which provided an updated perspective on emerging pathophysiological theories.

The trigeminal ganglion, which innervates cranial blood vessels, releases vasoactive neuropeptides such as SP, NKA, and CGRP. These substances are released from sensory fibers innervating blood vessels, inducing vasodilation and plasma protein extravasation via endothelial receptors, thereby promoting neurogenic inflammation. SP and NKA bind to endothelial receptors, causing endothelium-dependent vasodilation and increased vascular permeability, while CGRP acts on vascular smooth muscle receptors to induce vasodilation [52,53,54].

The dura mater, with its blood vessels innervated by trigeminal and upper cervical unmyelinated sensory fibers, is central to the trigeminovascular theory. Neurogenic plasma protein extravasation within the dura, triggered by SP or NKA, alters endothelial function, induces mast cell degranulation, and promotes platelet aggregation in and around post-capillary venules. These events lead to the release of chemical mediators—such as potassium, histamine, serotonin (5-HT), prostaglandins, leukotrienes, bradykinin, and neuropeptides—which contribute to pain generation [50,55,56].

Figure 2 shows an infographic on these topics.

Agonists of the 5-HT1D receptor, such as sumatriptan, inhibit plasma extravasation, platelet aggregation, and mast cell degranulation, thereby reducing inflammation. 5-HT1D and other 5-HT receptors located on unmyelinated C fibers play a crucial role in preventing plasma leakage. Various observations suggest that neurogenic inflammation originates from presynaptic terminals, as drugs that inhibit neuropeptide release via prejunctional mechanisms can block plasma extravasation in the dura mater and prevent subsequent endothelial changes, platelet aggregation, and mast cell activation [50].

Other agents—such as NSAIDs, opioids, and valproate—also reduce neurogenic inflammation by targeting traditional inflammatory mediators (e.g., potassium, histamine, prostaglandins, leukotrienes, bradykinin), as well as serotonin and neuropeptides secreted by sensory fibers. These drugs also act on additional receptors, including α2-adrenergic, H3, μ-opioid, and somatostatin receptors [57]. Blockade of neurogenic vascular leakage has also been demonstrated with migraine treatments such as NSAIDs [58], opioids [59], and valproate [60].

The primary mechanism responsible for neurogenic leakage from post-capillary venules in the dura involves SP acting via the NK1 receptor [61,62]. However, clinical trials targeting specific pathways—such as NK1 receptor antagonism or endothelin receptor blockade—have failed to demonstrate efficacy, casting doubt on their relevance in human migraine [48].

Imaging studies using gadolinium-enhanced MRI and SPECT have yielded inconsistent findings regarding dural extravasation during migraine attacks, leading to increased interest in neurogenic vasodilation. Experimental data show that SP, NKA, and CGRP induce vasodilation in dural vessels through selective stimulation of endothelial NK1 and CGRP receptors. Among these, CGRP causes prolonged vasodilation and is inhibited by 5-HT1B agonists in animal models, with 5-HT1D receptors serving a similar function in humans [48,63].

Elevated concentrations of SP and CGRP have been found in cerebral arteries compared to meningeal and temporal arteries. Studies on isolated human arteries—including meningeal, cerebral, and temporal vessels precontracted with prostaglandin F2α—have demonstrated that CGRP, SP, and NKA all induce vasodilation, with potency varying by artery type. In human cerebral arteries, neuropeptide Y (NPY) acts as a potent vasoconstrictor and enhances noradrenaline-induced contraction, whereas acetylcholine, peptide histidine methionine (PHM-29), and vasoactive intestinal peptide (VIP) are powerful vasodilators [63].

Numerous compounds and receptors have been investigated for their roles in migraine pathophysiology [48]. Key neuropeptides—such as CGRP, SP, and NKA—along with the more recently studied pituitary adenylate cyclase-activating peptide (PACAP), play central roles. PACAP, which shares structural similarity with VIP, also dilates cerebral arteries and enhances cerebral blood flow [64,65].

Table 1 reports a list of the main mediators and site and mechanism of action.

### 3.2. Mast Cells and Pro-Inflammatory Molecules

Mast cells, first linked to headache by Sicuteri in the 1960s [66], are now recognized as key contributors to the pathophysiology of headache. Upon degranulation, mast cells release a wide array of pro-inflammatory and pro-nociceptive mediators—including histamine, serotonin, TNF-α, IL-1, IL-6, leukotrienes, and prostanoids—that drive neuroinflammation [67]. Within the central nervous system (CNS), this process unfolds in the context of a unique immune microenvironment, where mast cells and microglia act as primary immune responders, and the glymphatic system facilitates immune surveillance despite the restrictive nature of the blood–brain barrier (BBB) [68].

Activated microglia release pro-inflammatory cytokines that recruit lymphocytes and activate astrocytes, amplifying the neuroinflammatory response. Although mast cells are scarce in the healthy brain, their numbers increase during neuroinflammation. They interact closely with glial cells and the cerebral vasculature through neuropeptides, cytokines, and adhesion molecules such as ICAM-1. Additionally, they are capable of direct granule transfer into neurons—a process known as transgranulation—which represents a novel mechanism of neuroimmune interaction [68].

Eftekhary et al. demonstrated that dural mast cells express CGRP receptor components, with co-localization of tryptase, calcitonin receptor-like receptor (CLR), and receptor activity-modifying protein 1 (RAMP1) in rats, but only CLR in human samples [69]. These mast cells, located abundantly along CGRP-positive fibers in the dura mater, play a critical role in migraine by releasing pro-inflammatory, vasodilatory, and neurosensitizing mediators [70,71,72].

Studies have shown that mast cell degranulation, triggered by CGRP, leads to the release of histamine and other mediators such as serotonin and prostaglandin I2. These agents sensitize meningeal nociceptors and contribute to migraine pain. Notably, brain-derived histamine is also released by neurons and endothelial cells. Activation of protease-activated receptor 2 (PAR2) further modulates nociceptive responses, and chronic depletion of mast cells has been shown to exacerbate sensitization [73,74,75]. Dural mast cell degranulation is associated with prolonged activation of the trigeminal pathway, as evidenced by increased pERK and c-Fos expression [76]. In animal models, nitroglycerin-induced migraine correlates with elevated IL-1β and IL-6 levels, indicating delayed meningeal inflammation [77].

Human studies have consistently reported elevated pro-inflammatory cytokines during migraine attacks. Neuropeptides such as CGRP and substance P stimulate T-cell activation, promoting the release of IL-1β, IL-6, and TNF-α, with SP eliciting a particularly strong TNF-α response [78,79]. These neuropeptides also modulate T-helper cell subsets. Cytokines, in turn, regulate CNS immunity by influencing microglial and macroglial activation, thereby sustaining neuroinflammation and immune modulation [80,81].

Since 1966, studies have documented immune dysregulation in migraine. Key findings include elevated histamine and TNF-α levels during and between attacks, increased IL-1β following attacks, and reduced interictal levels of IL-4, IL-2, and β-endorphin. While monocyte activity is generally impaired, transient increases during attacks have been observed. These immune changes—partially mediated by nitric oxide—have spurred interest in H3 histamine receptor agonists as potential therapeutic agents. No major alterations have been observed in immunoglobulins, complement factors, NK cells, or B lymphocytes [82].

Cytokine involvement in migraine has been extensively studied, particularly TNF-α and IL-1β. Perini et al. reported elevated levels of IL-10, TNF-α, and IL-1β during migraine attacks, with IL-10 remaining high throughout the acute phase. In contrast, interictal levels of TNF-α and IL-2 were lower compared to controls, supporting the presence of a fluctuating pro-inflammatory state in migraineurs [83].

Sarchielli et al. found transient increases in TNF-α, IL-6, IL-1β, and ICAM-1 during the early phases of migraine attacks, along with reduced expression of LFA-1 on T cells—indicating impaired leukocyte migration across the BBB. These findings support the existence of a transient pro-inflammatory state, potentially driven by CGRP, which facilitates immune cell activation and sustains neurogenic inflammation via modulation of the trigeminovascular system [84].

CGRP also promotes T-cell adhesion, facilitates leukocyte transmigration, and stimulates mononuclear cells to release TNF-α and IL-6, thereby increasing the expression of endothelial adhesion molecules. Additionally, it triggers mast cell degranulation, contributing to a complex neuroimmune network implicated in migraine pathogenesis [67,78,85].

Although genome-wide association studies (GWASs) have not identified significant single-gene associations in migraine, Gerring et al. found that pathway analyses highlight the involvement of immune–inflammatory processes. These include type I interferon responses, interferon signaling, microglial activity, NK cells, and monocyte-mediated mechanisms [86].

See Figure 3 for the relevant infographic.

### 3.3. Migraine Pathophysiology and Immune Involvement in Cortical Spreading Depression (CSD)

Migraine involves a combination of central and peripheral mechanisms. In approximately one-third of patients, it is preceded by aura, a phenomenon attributed to cortical spreading depression (CSD). CSD initiates the release of inflammatory mediators—including ATP, glutamate, calcitonin gene-related peptide (CGRP), and nitric oxide (NO)—which activate pial nociceptors and promote sustained activation of dural nociceptors. Furthermore, CSD stimulates immune responses: macrophages in the pia and dura exhibit prolonged activation, and dendritic cells demonstrate transient reductions in mobility, potentially recruiting T cells and extending the inflammatory response. This neuroimmune activation sensitizes meningeal nociceptors and contributes to migraine pain, even in the absence of prior meningeal inflammation [87,88,89].

The glymphatic system, a perivascular clearance pathway formed by astroglial cells, plays a crucial role in the removal of metabolic waste, proteins, and neurotransmitters from the central nervous system (CNS). Most active during sleep, its dysfunction has been implicated in various neurological disorders [90].

Schain et al. demonstrated in a mouse model that CSD impairs glymphatic function by inducing arterial constriction, delaying glymphatic flow, and reducing interstitial fluid clearance. This dysfunction hinders the removal of excitatory and inflammatory mediators such as glutamate and ATP, potentially prolonging cortical excitability and neuroinflammation. The resulting accumulation of inflammatory substances in the perivascular space may activate pial nociceptors, contributing to the onset of migraine and explaining the delayed headache that often follows aura [91].

Albrecht et al. (2019) provided the first in vivo evidence of neuroinflammation in migraine with aura using PET/MRI imaging and the glial marker [11C]PBR28 [92]. Their study revealed increased glial activation in key pain processing regions, including the periaqueductal gray, thalamus, and basal ganglia. Moreover, glial activity was positively correlated with migraine attack frequency, reinforcing the contribution of immune dysregulation to migraine pathophysiology [92,93,94].

Salahi et al. (2022) reviewed the immunological basis of migraine, emphasizing the role of trigeminovascular activation and neuroinflammation mediated by CGRP and pro-inflammatory cytokines such as IL-1β, TNF-α, and IL-6 [95]. Chronic glial activation within the trigeminal ganglia appears to facilitate pain sensitization and migraine chronification. Although cytokines play a central role, current evidence suggests that the blood–brain barrier remains largely intact during migraine attacks, pointing to a predominance of peripheral immune mechanisms. Genetic associations (e.g., MEF2D, NLRP3) and comorbidities with autoimmune diseases—including multiple sclerosis (MS), rheumatoid arthritis (RA), systemic lupus erythematosus (SLE), and psoriasis—further support an immunological component in migraine. Targeted therapies against CGRP, PACAP, and pro-inflammatory mediators, along with promising roles for vitamin D3 and dietary supplements, are currently under investigation [95].

Zhao (2024) explored the links between migraine risk factors, inflammatory proteins, and immune cell profiles [96]. The study reported associations between smoking, obesity, and altered expression of inflammatory markers such as IL-1α, CDCP1, TRAIL, and CD6. Nine proteins—including CD6 and beta-NGF—were positively associated with migraine, while others like CCL19 showed inverse correlations. Although correlations with 14 immune cell types were identified, none reached statistical significance. Comorbid conditions—especially anxiety, depression, and sleep apnea—showed strong associations with migraine risk, with anxiety demonstrating colocalization. However, no evidence was found for direct mediation via immune markers, suggesting that systemic inflammation, modulated by psychosocial and metabolic factors, may act through indirect pathways to influence migraine pathophysiology [96].

All of these concepts are presented in Figure 4.

### 3.4. Insulin Resistance and Neurogenic Inflammation

Insulin resistance (IR), a hallmark of type 2 diabetes mellitus (T2DM), involves impaired cellular responses to insulin in key metabolic tissues such as skeletal muscle, liver, and adipose tissue. Disruption of the insulin signaling cascade—particularly the phosphoinositide 3-kinase (PI3K)/AKT pathway—leads to a cascade of metabolic disturbances, including hyperglycemia, dyslipidemia, and chronic inflammation [97,98]. The interplay between oxidative stress, inflammation, and mitochondrial dysfunction lies at the core of IR and its associated comorbidities [98,99]. Pancreatic β-cells are central to glucose homeostasis, with insulin production regulated by transcription factors such as PDX-1, MafA, and NeuroD1 [100]. Chronic hyperglycemia and lipotoxicity induce endoplasmic reticulum (ER) stress, activating the unfolded protein response (UPR), which can lead to apoptosis when homeostasis is not restored. Additionally, pro-inflammatory cytokines such as IL-1β and TNF-α inhibit insulin gene transcription via the NF-κB pathway, further impairing β-cell function [98].

Mitochondrial dysfunction also impairs glucose-stimulated insulin secretion (GSIS). Reduced ATP production and altered mitochondrial dynamics contribute to β-cell failure. Therapies targeting mitochondrial biogenesis and reactive oxygen species (ROS) detoxification are under investigation [101]. Adipose tissue, an active endocrine organ, secretes adipokines that influence systemic metabolism. In obesity and IR, adipokine profiles are altered, with molecules such as CTRP3 (which enhances insulin sensitivity via AMPK activation) and WISP1 (which impairs insulin signaling through IRS-1 phosphorylation) playing opposing roles [102].

Chronic inflammation in adipose tissue—driven by macrophage infiltration and cytokine release (TNF-α, IL-6, CRP)—disrupts the IRS-1/PI3K/AKT axis. This inflammatory–metabolic crosstalk further aggravates insulin resistance, underscoring the relevance of anti-inflammatory interventions in T2DM [97]. An imbalance between ROS production and antioxidant defenses leads to oxidative stress, damaging cellular components and insulin receptors. Impaired oxidative phosphorylation (OXPHOS) and decreased mitochondrial biogenesis—regulated by PGC-1α—compromise energy homeostasis in insulin-sensitive tissues [98,101]. Novel therapeutic approaches include natural compounds and targeted molecular therapies. For instance, pomegranate peel extract improves hepatic insulin sensitivity by inhibiting NF-κB and reducing oxidative stress [99]. Curcumin acts via Nrf2 activation and TNF-α inhibition, improving systemic insulin action [102]. CGRP antagonists also reduce neurogenic inflammation, beneficial for T2DM-related complications such as neuropathy and retinopathy [103]. Lifestyle interventions remain essential. Caloric restriction (CR) activates AMPK and SIRT1, enhancing mitochondrial efficiency and reducing inflammation [98]. Aerobic exercise improves insulin sensitivity via PGC-1α and GLUT4 upregulation [97].

The brain is an insulin-sensitive organ, with widespread insulin receptor (InsR) expression affecting metabolism, mood, cognition, and synaptic function [104]. IR in the brain is associated with Alzheimer’s disease, depression, and schizophrenia. T2DM doubles the risk of depression and is a strong risk factor for mild cognitive impairment (MCI) and dementia. Schizophrenia patients have a 2–3 times higher prevalence of T2DM, only partly explained by antipsychotic use [105].

In neurons, insulin modulates dopaminergic signaling, especially in the ventral tegmental area (VTA), regulating motivation, food intake, and reward behavior. In astrocytes and microglia, insulin influences neurotransmission, synaptic plasticity, and immune responses. Brain endothelial cells mediate insulin transport across the blood–brain barrier (BBB), impacting central insulin sensitivity [105].

Insulin also modulates neurotransmitter systems, including GABA, NMDA, and AMPA receptors. Hippocampal IR impairs long-term potentiation (LTP), contributing to cognitive deficits [105].

Finally, insulin has a metabolic impact in the brain. Insulin signaling affects brain cholesterol metabolism, mitochondrial function, and oxidative stress. Brain insulin resistance is linked to reduced cholesterol synthesis, which may contribute to Alzheimer’s disease progression [105]. Insulin regulates neuroinflammation by modulating microglial activity. In conditions such as obesity and metabolic syndrome, microglia adopt a pro-inflammatory phenotype, releasing cytokines that exacerbate brain IR. In Alzheimer’s disease, brain IR reduces amyloid-β (Aβ) clearance and promotes neurodegeneration. Inflammatory cytokines (e.g., IL-6, TNF-α) impair brain insulin signaling, worsening cognitive outcomes. In animal models, intranasal insulin therapy improves synaptic plasticity, reduces neuroinflammation, and restores insulin sensitivity [105].

Table 2 gives a synthetic overview of this topic.

### 3.5. Migraine and Insulin Resistance: Clinical Evidence and Observations

Although the exact pathophysiology of migraine remains incompletely understood, growing evidence highlights a significant metabolic component, particularly related to insulin resistance (IR) and overall metabolic dysfunction.

The first study to explore altered insulin sensitivity in migraine patients was conducted by Rainero et al. in 2005 [106]. Since then, several studies have supported this association [104,107,108,109,110]., with the notable exception of the investigation by Sacco et al. in 2014 [111].

Sacco et al. assessed insulin resistance in patients with migraine with aura (MwA) and without aura (MwoA) to evaluate a possible association between migraine subtypes and IR. Their findings revealed significantly higher fasting glucose levels in MwA patients compared to MwoA and healthy controls (4.9 vs. 4.7 vs. 4.6 mmol/L; *p* = 0.018). Elevated glucose levels were associated with an increased risk of MwA compared to MwoA (OR = 5.32; *p* < 0.05), but not when compared to the control group. However, no significant correlations were found between insulin resistance and migraine frequency, duration, or severity. Notably, the study applied strict exclusion criteria, thereby eliminating participants with pre-existing conditions or symptoms frequently associated with IR [111].

In 2007, through a case–control design, we evaluated insulin and glucose dynamics in patients with migraine, individuals with other types of headache, and headache-free controls. Participants underwent an oral glucose tolerance test (OGTT), measuring blood glucose and insulin levels at baseline and at 30, 60, and 120 min following a 75 g glucose load. The results demonstrated that all headache groups exhibited higher post-load glucose levels compared to controls. However, migraine patients specifically showed a pronounced hyperinsulinemic response [112] (Figure 5A,B).

Following initial findings suggesting altered insulin sensitivity in migraine patients, several studies have explored the relationship between insulin resistance (IR), metabolic syndrome (MetS), and migraine, including the potential for targeted therapeutic interventions.

Bhoi et al. (2012) conducted a study in Southeast Asia examining the association between IR, MetS, and migraine [113]. Among migraine patients, 31.9% met the criteria for MetS and 11.1% had IR, while only 9.6% were classified as obese. However, central obesity was present in 57.8%. MetS was more common among older individuals, females, and those with longer migraine duration and multiple triggers. Notably, insulin resistance was significantly associated with the duration of migraine attacks but not with attack frequency, severity, or aura. Patients with MetS had more frequent triggers (e.g., head washing) and longer attack durations (2.0 ± 0.8 vs. 1.7 ± 0.8 days, *p* = 0.05), but no control group was included in the study [113].

Wang et al. (2017) carried out a case–control study in a Chinese community to investigate the link between glucose metabolism, IR, and migraine [114]. In a subset of 181 participants who underwent OGTT, no significant differences in HbA1c, HOMA-IR, HOMA-B, or QUICKI were observed between migraineurs and controls overall. However, migraineurs with prediabetes had significantly higher fasting insulin, HOMA-IR, and lower QUICKI values compared to prediabetic controls. Interestingly, diabetes mellitus (DM) was negatively associated with migraine in the OGTT subgroup, though this finding was not replicated in the full cohort of 1832 individuals. The authors hypothesized that chronic hyperglycemia in DM may lead to metabolic adaptations that reduce migraine susceptibility [114].

A systematic review by Hosseinpour et al. (2021) similarly reported a reduced incidence of migraine among diabetic patients, possibly due to altered central pain processing in chronic hyperglycemia [115].

Ali et al. (2022) conducted a case–control study that further confirmed higher IR in migraine patients compared to non-migraine controls [116]. Migraine patients with IR experienced more frequent and severe attacks, as indicated by elevated scores on the Migraine Severity Scale (MIGSEV) and Headache Impact Test-6 (HIT-6). They also exhibited higher waist circumference, insulin levels, and HOMA-IR values (all *p* < 0.05). Interestingly, no significant differences in fasting glucose, triglycerides, or HDL cholesterol were found between groups. Moreover, MetS did not affect migraine severity or duration, while HOMA-IR values correlated positively with both severity and disability (*p* < 0.001) [117].

Additional support comes from the studies by Fava et al. and Gur-Ozmen et al. (2019) [118,119]. Fava et al. found a correlation between elevated fasting insulin and migraine intensity, particularly in patients with obesity or MetS [118]. Gur-Ozmen et al. highlighted the synergistic effect of obesity and IR in worsening migraine. Their study also examined the role of chronic medication use in developing IR among migraineurs. Factors associated with IR included central obesity, the use of metoclopramide during attacks, a family history of diabetes, and frequent NSAID use [119].

### 3.6. Mechanistic Pathways Linking Hyperinsulinism and Migraine

The interplay between hyperinsulinism, insulin resistance, and migraine offers emerging insights into migraine pathophysiology and potential therapeutic strategies. Hyperinsulinism, characterized by elevated circulating insulin levels, has been implicated in several mechanisms contributing to migraine onset and chronicity. The relationship between dysregulated insulin metabolism and migraine involves several interconnected pathways: TRPV1 sensitization and CGRP release, mitochondrial dysfunction with oxidative stress, and systemic and neurogenic inflammation.

Insulin modulates the trigeminovascular system by sensitizing transient receptor potential vanilloid 1 (TRPV1) receptors, thereby enhancing the release of calcitonin gene-related peptide (CGRP). In a key study, Rosta et al. [117] demonstrated that insulin sensitizes TRPV1 receptors in the trigeminovascular pathway, amplifying nociceptive signaling and increasing CGRP release.

Hyperinsulinism is closely associated with increased oxidative stress, which can lower the threshold for cortical spreading depression (CSD)—the electrophysiological phenomenon underlying migraine aura. Grinberg et al. (2017) [120] showed that hyperinsulinemia and related metabolic disturbances exacerbate neuronal excitability and impair mitochondrial function, thus promoting CSD and migraine onset.

The inflammatory consequences of hyperinsulinism have been supported by Ali et al. (2022) [116], who reported significantly elevated levels of pro-inflammatory cytokines (e.g., TNF-α, IL-6) in migraine patients with insulin resistance. This association underlines the role of systemic and neurogenic inflammation in linking metabolic dysfunction to migraine severity.

These mechanistic links offer potential avenues for targeted migraine treatment, especially in patients with metabolic comorbidities.

An exploratory study our research group conducted in 2017 [121] showed a 53% reduction in migraine frequency after 2 months of ALA supplementation in hyperinsulinemic patients. ALA, a mitochondrial antioxidant, improves insulin sensitivity and reduces oxidative stress, addressing two critical aspects of migraine pathophysiology.

A recent study by Olivito et al. (2024) [122] reported that an 8-week Mediterranean ketogenic diet reduced both migraine frequency and insulin resistance in patients with chronic migraine. The diet’s anti-inflammatory and insulin-sensitizing properties were instrumental in its therapeutic efficacy.

Halloum et al. (2024) [123] highlighted the role of glucagon-like peptide-1 (GLP-1) receptor agonists in reducing migraine frequency. These agents exhibit dual action by improving metabolic regulation and modulating nociceptive pathways, showing promise in metabolically driven migraine phenotypes.

Personalized treatment approaches for migraine could benefit from biomarker-driven strategies: insulin and glucose markers, oxidative stress biomarkers, and genetic associations. Fasting insulin levels and OGTT profiles have been proposed as screening tools to identify hyperinsulinemic migraineurs, as demonstrated by Ali et al. (2022) [116] and Cavestro et al. (2007) [112]. Grinberg et al. (2017) [120] recommended tracking malondialdehyde (MDA) and related oxidative stress markers to monitor disease activity and response to antioxidants. Zhang et al. (2024) [124], using Mendelian randomization, identified genes involved in insulin signaling pathways as potential therapeutic targets in migraine management.

Figure 6a,b represent these concepts.

### 3.7. Dietary and Nutritional Interventions for Migraine Management

A growing body of evidence supports the role of targeted dietary interventions in reducing the frequency, duration, and severity of migraine attacks. These interventions address both the metabolic and neuroinflammatory aspects of migraine pathophysiology. This section provides a comprehensive overview of the key mechanistic pathways influenced by diet and categorizes evidence-based strategies for nutritional management of migraine.

Diet influences the release of pro-inflammatory cytokines and free radicals, contributing to neurogenic inflammation. Dietary components rich in antioxidants and anti-inflammatory compounds (e.g., polyphenols, omega-3s) may counteract these processes. Fluctuations in blood glucose—particularly hypoglycemia and reactive hyperinsulinemia—can act as potent migraine triggers. Diets that stabilize glycemic control may prevent glucose-induced neuronal excitability. Nutrients such as sodium, potassium, and omega-3/omega-6 fatty acids influence vascular tone and nitric oxide production. Diet-induced endothelial dysfunction may contribute to migraine pathogenesis via impaired cerebral blood flow regulation. Migraineurs often exhibit impaired mitochondrial energy metabolism. Nutritional strategies targeting mitochondrial biogenesis and function (e.g., riboflavin, coenzyme Q10, magnesium) may improve cellular energy production and reduce attack susceptibility. Alterations in gut microbiota composition and increased intestinal permeability (leaky gut) may exacerbate systemic and neurogenic inflammation. Diets promoting gut health—such as fiber-rich or probiotic-rich diets—may play a modulatory role in migraine prevention.

### 3.8. Weight Loss

Obesity is a well-established risk factor for the chronification of migraine, contributing to the progression from episodic to chronic forms of the disorder [125]. The metabolic consequences of obesity—particularly insulin resistance, hypothalamic dysfunction, and impaired mitochondrial metabolism—are increasingly recognized as playing central roles in migraine pathophysiology. These associations have led to the exploration of weight loss interventions, both surgical and non-surgical, as potential strategies to reduce the burden of migraine.

A comprehensive meta-analysis conducted by Di Vincenzo et al. (2020) assessed the effect of weight reduction on several migraine-related outcomes, including attack frequency, pain intensity, disability, and duration, while also exploring potential underlying mechanisms [126]. This analysis, which included 10 studies with a total of 473 participants, demonstrated significant improvements in all measured parameters following weight loss. Specifically, the findings revealed a substantial reduction in monthly headache days (effect size [ES] = −0.78; *p* < 0.0001), a large reduction in pain intensity (ES = −1.04; *p* < 0.0001), a moderate reduction in disability scores (ES = −0.68; *p* < 0.0001), and a small but statistically significant decrease in attack duration (ES = −0.25; *p* = 0.017). These results underscore the role of weight loss as an effective non-pharmacological intervention for managing migraine.

The observed clinical benefits appear to be mediated by several interconnected metabolic and inflammatory mechanisms. First, obesity is associated with a state of chronic low-grade systemic inflammation, characterized by elevated levels of pro-inflammatory cytokines, including tumor necrosis factor-alpha (TNF-α), interleukin-6 (IL-6), C-reactive protein (CRP), and resistin. These cytokines contribute to migraine pathogenesis by promoting trigeminovascular activation and increasing calcitonin gene-related peptide (CGRP) release. Weight loss has been shown to reverse this inflammatory profile, decreasing circulating pro-inflammatory markers and increasing levels of adiponectin, an anti-inflammatory adipokine with potential neuroprotective effects [126]. Clinically, this anti-inflammatory shift may help suppress neurogenic inflammation, thereby reducing migraine susceptibility.

Improved insulin sensitivity represents another critical mechanism. Insulin resistance is prevalent among migraine patients and has been linked to hypothalamic dysregulation and impaired cerebral glucose metabolism. Although the meta-analysis did not uniformly report indices such as HOMA-IR or fasting insulin, several included studies noted improvements in glucose homeostasis following weight loss. Since hyperinsulinemia is known to promote vasodilation and CGRP-mediated neuroinflammatory responses, restoring insulin sensitivity may normalize vascular reactivity and reduce migraine frequency. These findings are particularly relevant for populations with metabolic syndrome or polycystic ovary syndrome (PCOS), where insulin resistance is frequently observed [126].

In addition to metabolic and inflammatory factors, weight loss may modulate hypothalamic neuropeptides involved in appetite regulation and migraine. Caloric restriction and subsequent weight loss have been associated with increased orexin levels—known to influence wakefulness and nociceptive modulation—and decreased leptin levels, which are often elevated in obesity and have pro-inflammatory properties. Ghrelin, another appetite-regulating hormone with neuroprotective potential, is often dysregulated in migraine. The reduction in leptin resistance following weight loss may attenuate central neuroinflammation and reduce the likelihood of migraine chronification [126].

Mitochondrial dysfunction has also been implicated in migraine, with evidence suggesting impaired cortical energy metabolism and reduced ATP production. The ketogenic diets—characterized by a high-fat, low-carbohydrate intake—has shown particular promise in this context, owing to its capacity to enhance mitochondrial efficiency and induce ketone body production (e.g., β-hydroxybutyrate), which serves as an alternative energy substrate for neurons. According to Di Vincenzo et al., ketogenic dietary interventions demonstrated the strongest effect size among the various weight loss strategies assessed, likely due to their role in enhancing mitochondrial metabolism, reducing oxidative stress, and stabilizing neuronal excitability [126].

Table 3 compare some dietary weight loss strategies and their metabolic effects.

These findings suggest that the metabolic benefits of weight loss, rather than absolute weight reduction, drive migraine relief. The meta-analysis leaves open some unanswered questions: What is the optimal dietary composition for migraine management (e.g., ketogenic vs. anti-inflammatory diets)? What is the threshold of weight loss required for sustained migraine relief? How does gut microbiome modulation following weight loss influence migraine susceptibility? Can combination therapy (weight loss + pharmacological treatment) produce synergistic effects? The author concludes that given the strong metabolic component of migraine pathophysiology, future research should focus on precision nutrition approaches, leveraging individualized metabolic profiling to optimize dietary and lifestyle interventions for migraine prevention.

### 3.9. Diets

Gazerani has published two comprehensive reviews on the interplay between migraine and diet. In the first, the author explores the bidirectional relationship, emphasizing that while dietary factors can influence the frequency and severity of migraine attacks, the condition itself may, in turn, modulate eating behaviors and nutritional choices [127]. This interaction is underpinned by evidence linking migraine to mitochondrial dysfunction and cerebral energy deficits, prompting the hypothesis that targeted dietary interventions may assist in migraine management. Traditional strategies have predominantly focused on the identification and elimination of common dietary triggers—such as caffeine, alcohol, monosodium glutamate (MSG), and tyramine-rich foods. However, emerging perspectives underscore the importance of dietary consistency, nutritional quality, and the therapeutic potential of functional food components. Recent interest has grown in the use of nutraceuticals and micronutrient supplementation for migraine prevention, particularly omega-3 fatty acids, magnesium, riboflavin (vitamin B2), and coenzyme Q10. Moreover, the gut–brain axis has garnered attention as a potential modulator of migraine pathophysiology. Alterations in the gut microbiota of migraineurs suggest that probiotics and prebiotics could serve as adjunctive therapeutic agents. The association between obesity and increased migraine frequency further supports the role of metabolic interventions, with evidence suggesting that weight reduction—via low-calorie or ketogenic dietary regimens—may yield clinical benefit. These findings reinforce the concept of a dynamic and reciprocal relationship between diet and migraine. Gazerani proposes that future research should prioritize individualized nutritional strategies, with a focus on the gut–brain–migraine interface [127].

In a more recent review published in 2023, Gazerani revisited and expanded upon these themes, incorporating updated evidence on dietary triggers, effective interventions, and innovative therapeutic models [128]. This review highlights that while alcohol and caffeine remain the most consistently reported triggers in the literature, other commonly suspected foods—such as chocolate, cheese, and wine—may function as early prodromal cues rather than direct migraine inducers. Among the dietary approaches with therapeutic potential, the ketogenic diet (KD) is noted for its ability to reduce both the frequency and intensity of attacks, although gastrointestinal side effects and long-term adherence remain challenging. The DASH (Dietary Approaches to Stop Hypertension) diet has shown promise in mitigating migraine severity, particularly in women, likely through its effects on vascular tone and systemic inflammation. Personalized low-glycemic diets have also demonstrated efficacy in reducing the number of headache days, especially when tailored using real-time metabolic data.

A key point of emphasis is the balance between omega-3 and omega-6 fatty acids: increasing anti-inflammatory omega-3 intake while reducing pro-inflammatory omega-6 sources has been associated with a lower headache burden. In terms of microbiome-targeted therapy, the review notes that although migraine patients often exhibit altered gut microbiota, current meta-analyses do not support a consistent benefit from probiotic supplementation, indicating a need for more refined, individualized approaches.

This line of inquiry is supported by the recent large-scale randomized controlled trial conducted by Evers et al. (2025), which evaluated the efficacy of the digital therapeutic platform *sinCephalea* [129]. This system utilizes continuous glucose monitoring (CGM) to guide users toward a personalized low-glycemic dietary pattern. In a cohort of over 800 patients with episodic migraine, the *sinCephalea* intervention yielded a statistically significant reduction in monthly migraine days when compared to standard care. The therapeutic effect was particularly pronounced in participants with high adherence to dietary recommendations. Additionally, *sinCephalea* users demonstrated improvements in migraine-related disability scores, including the HIT-6 and MIDAS instruments, with no adverse events attributable to the intervention. These findings illustrate the emerging role of digital health technologies in delivering precision nutrition, offering a scalable and individualized approach to migraine prevention. See Table 4 for comparison between Gazerani and Evers results.

### 3.10. Anti-Inflammatory, Mediterranean, and DASH Diets

An increasing body of research supports the role of anti-inflammatory dietary patterns in the prevention and management of migraine. These diets—particularly the Mediterranean diet (MD) and the DASH (Dietary Approaches to Stop Hypertension) diet—have shown favorable effects on systemic inflammation, neurovascular function, and metabolic health, all of which are implicated in migraine pathophysiology.

Several studies have highlighted the association between dietary inflammatory load and migraine frequency or severity. Ghoreishy et al. demonstrated that a higher Dietary Inflammatory Index (DII), indicative of a pro-inflammatory diet, was positively correlated with increased migraine frequency and severity. Conversely, anti-inflammatory diets rich in fruits, vegetables, nuts, and omega-3 fatty acids were associated with fewer attacks [130]. Supporting this, Hajishizari et al. showed that a higher Dietary Antioxidant Quality Score (DAQS) was significantly associated with lower headache frequency, reduced migraine intensity (measured with the Visual Analog Scale), and decreased disability (measured by MIDAS scores) [131]. Nutrient-specific findings suggest that vitamin C and vitamin E exert a protective role, reducing pain intensity and attack frequency, respectively, likely through antioxidant and neuroprotective mechanisms.

The Mediterranean diet (MedDiet) and the DASH diet are among the most studied dietary interventions for chronic diseases and are increasingly recognized for their role in migraine prevention. The MedDiet, characterized by a high intake of monounsaturated fats (olive oil), omega-3-rich fish, polyphenol-rich plant foods, and moderate wine consumption, exerts broad anti-inflammatory and metabolic effects [132,133,134]. In the study by Arab et al. [132], adherence to the MedDiet significantly improved insulin sensitivity and reduced systemic inflammatory markers. Similarly, the DASH diet, with its emphasis on whole grains, fruits, vegetables, and low-fat dairy, has shown significant benefits in metabolic syndrome and migraine. Arab et al. [135] and Hajjarzadeh et al. [136] reported improved fasting glucose levels and a marked reduction in inflammatory cytokines such as IL-6 and TNF-α. Furthermore, the DASH diet was associated with a reduction in migraine frequency and severity, alongside improvements in psychological well-being.

Both the MedDiet and the DASH diet influence key migraine pathways. Polyphenols from plant-based foods may modulate the release of calcitonin gene-related peptide (CGRP), a key neuropeptide involved in migraine attacks. High-fiber intake fosters a diverse gut microbiome, enhancing short-chain fatty acid (SCFA) production, which is known to dampen systemic and neurogenic inflammation [130,137]. Moreover, nitrate-rich vegetables (e.g., leafy greens and beets), common in both diets, enhance nitric oxide synthesis, supporting endothelial function and reducing vasospasm events [136].

The study by Tirani et al. [138] explored the association between dietary phytochemical intake and metabolic health, focusing on serum levels of adropin and BDNF—key biomarkers of neuroprotection and energy metabolism. A higher Dietary Phytochemical Index (DPI) was associated with improved metabolic profiles, potentially offering neuroprotective benefits in migraine patients.

Additional dietary components have been explored in relation to migraine. These include magnesium, curcumin, coenzyme Q10, riboflavin, folate, zinc, tryptophan, and vitamin B1—all of which have demonstrated modulatory effects on oxidative stress, inflammation, and mitochondrial function [34,137,138,139,140,141,142,143,144,145,146,147,148,149,150,151,152,153,154,155,156]. Dietary quality and diversity have also been positively correlated with migraine outcomes, as shown through the Dietary Diversity Score (DDS) and studies on diet quality indices [141,142,147].

Collectively, these findings underscore the therapeutic potential of anti-inflammatory, nutrient-rich diets in migraine management. The Mediterranean and DASH diets, in particular, stand out due to their multidimensional effects on metabolic, vascular, and neuroinflammatory pathways. The integration of personalized nutritional strategies, possibly aided by digital health tools, represents a promising frontier in the non-pharmacological treatment of migraine.

Emerging evidence underscores the critical role of nutrition in modulating the pathophysiology and clinical course of migraine. Multiple dietary components—both protective and deleterious—have been implicated in the onset, frequency, and severity of migraine attacks, with particular emphasis on inflammation, oxidative stress, mitochondrial function, and vascular regulation.

Pro-inflammatory diets, particularly those aligned with Western dietary patterns rich in processed foods, trans fats, refined sugars, and omega-6 fatty acids, have consistently been associated with increased migraine prevalence and severity. These diets typically exhibit high scores on the Dietary Inflammatory Index (DII), reflecting their potential to exacerbate systemic and neurogenic inflammation. This low-grade chronic inflammation contributes to sensitization of the trigeminovascular system, oxidative stress, and endothelial dysfunction—factors that are central to migraine pathophysiology. Studies have shown that frequent consumption of red meat, processed meats, hydrogenated oils, and high-sugar foods correlates with a higher incidence of migraine attacks and chronic migraine progression [130,140,141].

In contrast, anti-inflammatory dietary patterns such as the Mediterranean and DASH diets have demonstrated protective effects against migraine. These diets are characterized by high intakes of vegetables, fruits, whole grains, legumes, and omega-3-rich sources such as fish and nuts. They are also low in processed foods and saturated fats, and are abundant in antioxidants and polyphenols. Data from observational and interventional studies indicate that higher adherence to these diets is associated with lower migraine frequency, reduced severity, and improved quality of life. For example, Hajishizari et al. [131] found that higher Dietary Antioxidant Quality Scores (DAQSs) were inversely correlated with migraine intensity, disability (as measured by MIDAS), and headache days per month. Notably, vitamin C and E intake showed a negative association with migraine severity and frequency, suggesting a neuroprotective and anti-inflammatory of for these micronutrients.

Furthermore, omega-3 fatty acids, particularly eicosapentaenoic acid (EPA) and docosahexaenoic acid (DHA), have shown promise as adjunctive therapies in migraine management. Their anti-inflammatory properties stem from their ability to reduce the synthesis of pro-inflammatory eicosanoids and promote the production of specialized pro-resolving mediators such as resolvins and protectins. Clinical evidence indicates that increasing dietary omega-3s—especially when coupled with a reduction in omega-6 fatty acid intake (H3-L6 model)—can significantly reduce the number of monthly migraine days, decrease pain intensity, and lower reliance on acute medications [128,132,133]. This dietary modulation also impacts CGRP signaling, a key neuropeptide involved in migraine pathogenesis, offering a mechanistic rationale for these benefits.

In this context, the important meta-analysis conducted by García-Pérez-de-Sevilla and González-de-la-Flor (2025) provides significant insights into the efficacy of fatty acid supplementation in the management of migraine [157]. By analyzing six randomized controlled trials (n = 407 patients), the authors found that omega-3 supplementation—specifically eicosapentaenoic acid (EPA) and docosahexaenoic acid (DHA), often combined with omega-6 restriction—resulted in clinically relevant benefits. These included reductions in pain intensity (standardized mean difference [SMD] = −1.77), attack duration (SMD = −0.77), episode frequency (SMD = −1.91), and migraine-related disability, as measured by the Headache Impact Test (HIT-6) score (SMD = −2.44). Particularly promising were the effects of alpha-lipoic acid, a potent antioxidant known for its endothelial protective properties, which showed the most pronounced improvements across clinical outcomes [157]. The therapeutic action of omega-3 fatty acids appears to be mediated through the downregulation of pro-inflammatory cytokines, the enhancement in antinociceptive mediators such as resolvins and protectins, and the modulation of the endocannabinoid system. Notably, the combined strategy of increasing omega-3 intake while reducing omega-6 fatty acids demonstrated a synergistic effect, suggesting that the balance between these two lipid classes plays a critical role in migraine pathophysiology [157]. These findings reinforce the growing body of evidence supporting anti-inflammatory dietary approaches in migraine management and pave the way for more precise nutritional recommendations, particularly in patients with chronic migraine and metabolic comorbidities. Nevertheless, larger and longer-term randomized controlled trials are needed to confirm these outcomes and to standardize dietary protocols.

In addition to general dietary patterns, specific micronutrients have demonstrated efficacy in reducing migraine burden. Riboflavin (vitamin B2) has been well studied in this regard, with high doses (≥400 mg/day) shown to decrease attack frequency, possibly through enhancements in mitochondrial energy metabolism. Thiamine (vitamin B1) has also been implicated, particularly in women and older adults, with observational data supporting its inverse association with migraine prevalence [156]. Supplementation with coenzyme Q10—an essential component of the mitochondrial electron transport chain—has led to significant reductions in attack frequency and severity, especially in patients with documented mitochondrial dysfunction or fatigue. Curcumin, a polyphenol derived from turmeric, exhibits anti-inflammatory and antioxidant effects through modulation of NF-κB and Nrf2 pathways, which are increasingly being explored in migraine trials [150,151].

Magnesium is another key nutrient, particularly effective in patients with menstrual migraine or aura. It plays a central role in regulating neuronal excitability and preventing cortical spreading depression (CSD), a wave of depolarization thought to underlie migraine aura and pain generation. Clinical guidelines often recommend magnesium supplementation (in the form of magnesium citrate or glycinate) as a preventive option due to its favorable safety profile [128,145].

Finally, antioxidant micronutrients such as vitamin C and zinc have been highlighted for their roles in mitigating oxidative stress and inflammation in the central nervous system. These nutrients have shown inverse associations with migraine prevalence and severity in population-based studies, supporting their inclusion in comprehensive dietary strategies for migraine prevention [131,143,155].

Taken together, these findings reinforce the role of targeted dietary modifications in migraine prevention. Nutritional interventions, particularly those that reduce dietary inflammation, optimize mitochondrial function, and restore neurovascular balance, may offer effective, non-pharmacological adjuncts in the management of this disabling neurological disorder. Further randomized controlled trials are warranted to clarify dose–response relationships and refine patient stratification for personalized dietary therapies.

### 3.11. Ketogenic Diet/Low-Carb Diet and Migraine

The ketogenic diet (KD), originally developed for epilepsy management, has emerged as a promising non-pharmacological strategy for migraine prevention. In recent years, a growing body of evidence has supported its role in modulating migraine pathophysiology through diverse metabolic and neuroinflammatory pathways.

Finelli et al. (2023) provided a comprehensive overview of chronic primary headaches and highlighted the potential of combining dietary interventions—particularly the ketogenic diet—with conventional treatments [158]. The diet’s hallmark feature is the induction of ketosis, a metabolic state wherein the body utilizes ketone bodies (e.g., β-hydroxybutyrate, acetoacetate) as its primary energy substrate instead of glucose. This shift has several implications for the neurological system, particularly in migraineurs.

One of the primary mechanisms through which the KD exerts its effects is by improving neuronal energy metabolism. Migraine has been associated with mitochondrial dysfunction and impaired ATP production; the KD may help restore metabolic efficiency and cellular energy homeostasis [159,160]. Moreover, the KD reduces oxidative stress and downregulates pro-inflammatory cytokines such as TNF-α and NF-κB, both of which are implicated in migraine pathogenesis [158,160,161].

Ketosis also appears to influence neurotransmitter balance. It enhances GABAergic signaling and dampens excessive glutamatergic activity, thus modulating cortical excitability—a key factor in migraine generation [36,162]. In addition, KD may suppress trigeminovascular activation through inhibition of calcitonin gene-related peptide (CGRP), a neuropeptide central to migraine pain transmission [158].

Clinical studies further support the diet’s efficacy. Di Lorenzo et al. documented that in two migraine-prone twin sisters, attacks ceased within three days of achieving ketosis and recurred after cessation, independent of weight loss—suggesting a direct neuroprotective role of ketone bodies [159]. In a subsequent prospective study comparing 96 overweight migraine patients, those on the KD experienced a rapid reduction in attack frequency within one month, while those on a standard low-calorie diet improved more gradually. Upon switching back from the KD to the standard diet, migraines worsened again, reinforcing the metabolic specificity of the effect [115].

The same authors later evaluated cortical responses through evoked potential studies, showing that migraineurs had abnormally heightened cortical reactivity and impaired habituation. These neurophysiological markers normalized after the KD, indicating improved synaptic plasticity and suggesting a cortical site of action [36].

A systematic review by Caminha et al. (2022) analyzing 10 clinical trials found that the KD consistently reduced migraine frequency and severity with minimal side effects. However, they noted limitations such as small sample sizes and heterogeneity in study designs [160]. Similarly, Barbanti et al. reviewed data from 150 patients and found that migraine relief often began within days of initiating the KD. Notably, these benefits were independent of weight loss, underscoring the primary neurophysiological rather than metabolic mechanism of action [161].

Tereshko et al. compared three KD protocols—a standard KD, a very-low-calorie ketogenic diet (VLCKD), and a low-glycemic index diet (LGID)—in a pilot study of 76 migraineurs. All approaches significantly reduced headache intensity, frequency, and disability. Fatigue, a common migraine comorbidity, also improved across protocols. The VLCKD yielded the strongest clinical benefits, likely due to greater ketone production [162].

Despite promising outcomes, several limitations warrant caution. Short intervention durations (usually ≤6 months), variable macronutrient compositions, and a lack of pediatric data restrict generalizability. Gastrointestinal side effects and strict dietary adherence are potential barriers, particularly in long-term settings [160,161].

Nevertheless, the KD represents a compelling option for specific patient populations. Ideal candidates may include overweight or obese individuals with metabolic dysregulation, patients with drug-resistant migraine, or those with coexisting neurological disorders like epilepsy. Clinical implementation should include regular monitoring of lipid profiles, renal function, and micronutrient levels [158,160].

Interestingly, publications have reported that the KD leads to a ≥50% reduction in migraine frequency in drug-resistant patients. Improvements in attack intensity and duration were also observed, especially among those with aura. Higher circulating β-hydroxybutyrate levels correlated with better outcomes [159,163,164]. In particular, the VLCKD showed rapid efficacy in obesity-associated migraine, suggesting that insulin resistance may underlie the migraine phenotype in some patients [165,166]. Similarly, low-glycemic index diets help stabilize blood glucose and may reduce hypoglycemia-induced headaches [146,167].

The mechanistic underpinnings of the KD’s benefits are multifactorial. Key effects include the enhancement in GABAergic over glutamatergic activity, improved mitochondrial ATP production by bypassing dysfunctional complexes, and the suppression of neurogenic inflammation via reduced TNF-α and IL-1β expression [158,160,161].

In summary, the ketogenic diet offers a promising, metabolically targeted intervention for migraine management. Its benefits span anti-inflammatory effects, neurochemical modulation, and improved mitochondrial function. While further large-scale randomized controlled trials are essential to validate these findings and define optimal protocols, current evidence suggests that the KD may serve as a valuable tool—especially for patients with refractory migraine or metabolic comorbidities.

### 3.12. The Mediterranean Diet May Offer an Alternative Approach

In this context, the observational study by Arab et al. (2023) [132] provides further evidence supporting the role of anti-inflammatory dietary patterns in migraine management. Their findings demonstrated that adherence to the Mediterranean diet was significantly associated with reductions in both migraine frequency and severity. The underlying mechanisms are likely related to the anti-inflammatory properties of the diet and its capacity to stabilize glucose levels. Given its palatability and flexible structure, the Mediterranean diet may serve as a viable alternative for patients who struggle with the more restrictive ketogenic diet (KD).

Building upon this concept, and following the identification of hyperinsulinism as a specific metabolic hallmark in migraine patients [118], my research group developed and implemented a modified version of the Mediterranean diet tailored to individuals with insulin-related dysregulation. This adapted dietary plan—termed the “fractionated diet”—has been utilized in our clinical practice since 2005 and emphasizes a structured, low-glycemic, and frequent meal distribution approach. Key components of the fractionated diet include the avoidance of added sugars and fasting periods longer than three hours during waking hours, the incorporation of protein at every meal (at least one full portion during main meals and half portions during snacks), the restriction of complex carbohydrates to small portions, and the consumption of fruit only on a full stomach. Alcohol intake is discouraged, particularly beer and spirits, with small quantities of wine permitted only with meals. When formally calculated, this regimen provides approximately 45% of total caloric intake from carbohydrates.

An observational study was conducted in 2005 to evaluate the effects of this dietary intervention on migraine in patients exhibiting altered glucose–insulin metabolism [168]. The study population consisted of 173 consecutively enrolled migraine patients (138 women and 35 men) who were assessed over a 12-month period at our hospital’s headache clinic. Each patient underwent baseline laboratory testing, including a standard oral glucose tolerance test (OGTT) following a 12 h fast, with measurements of both glucose and insulin levels. A fractionated low-carbohydrate diet was then recommended to those with metabolic abnormalities, and follow-up OGTTs were performed after 3 to 6 months. Among the cohort, 132 patients (102 women and 30 men) demonstrated glucose–insulin dysfunction, while 41 had normal metabolic profiles. A total of 64 patients completed the follow-up period. The headache severity index—calculated as the product of the number of monthly headache days and severity per episode on a scale of 1 (mild) to 3 (severe)—decreased significantly from 31 ± 28 (SD) to 12 ± 21 (SD) (*p* < 0.0001) following the dietary intervention. Fasting glucose and post-load OGTT glucose levels showed a modest, non-significant reduction (pre-diet: FBG 96 mg/dL; 30 min: 143; 60 min: 113; 120 min: 93 vs. post-diet: FBG 99; 30 min: 131; 60 min: 104; 120 min: 83; *p* < 0.3). In contrast, insulin levels demonstrated a significant decline across all OGTT time points (pre-diet: FBG 10 µU/mL; 30 min: 100; 60 min: 99; 120 min: 55 vs. post-diet: FBG 7; 30 min: 74; 60 min: 75; 120 min: 38; *p* < 0.1).

These results suggest that a fractionated low-carbohydrate diet may serve as an effective non-pharmacological strategy in the management of migraine, particularly in patients with underlying insulin resistance or hyperinsulinism. Notably, the clinical improvements in headache severity appeared to correlate more closely with reductions in insulin levels than with changes in glucose, both under fasting and postprandial conditions. This supports the hypothesis that insulin dysregulation plays a pivotal role in migraine pathogenesis.

Furthermore, these findings align with the emerging literature that implicates insulin in central nervous system function, reinforcing the potential utility of targeted dietary interventions in modulating both metabolic and neurological outcomes in migraine patients. Continued research through larger, controlled trials is warranted to validate these results and refine nutritional protocols for this patient population.

### 3.13. Do Not Skip Meals

Growing evidence suggests that irregular eating habits, including meal skipping and prolonged fasting, may act as significant yet underrecognized triggers for migraine attacks. The scoping review by Legesse et al. (2025) comprehensively examined 36 studies encompassing a wide range of populations and research designs, highlighting a robust association between erratic meal timing and increased migraine frequency and severity [169]. The pathophysiological basis for this association appears to involve several interrelated mechanisms. Chief among them is the induction of hypoglycemia, which can destabilize neuronal homeostasis and increase cortical excitability—an established precursor of cortical spreading depression, particularly in migraine with aura. Hypoglycemia may also precipitate compensatory hormonal responses, such as elevated cortisol and catecholamine release, further sensitizing trigeminovascular pathways. Additionally, irregular meals may exacerbate neurogenic inflammation and promote the release of pro-inflammatory mediators, including nitric oxide (NO), calcitonin gene-related peptide (CGRP), and other vasoactive substances implicated in migraine attacks. Disruption of metabolic rhythms may also alter autonomic tone and ion channel activity, further compounding susceptibility in predisposed individuals. Clinical observations have consistently reported that skipping breakfast, in particular, is a common prodrome or direct precipitant in patients with chronic migraine. Migraine subtypes—including menstrual, retinal, and abdominal migraines—seem particularly sensitive to glycemic fluctuations, suggesting that glycemic stability may be a shared therapeutic target across phenotypes. Notably, the review also identifies meal regularity as a potentially protective behavioral intervention. Patients adhering to structured eating schedules appear to experience fewer migraine episodes and report lower symptom burden, although high-quality randomized controlled trials are still needed to confirm causality. Despite some inconsistencies in the literature—largely attributable to study design limitations such as small sample sizes, retrospective recall bias, and heterogeneity in dietary assessment tools—the collective findings underscore the importance of dietary rhythm in migraine management. In conclusion, regular, nutritionally balanced meals should be recommended as a first-line lifestyle modification in migraine prophylaxis. These findings reinforce the role of metabolic homeostasis in neurovascular health and support the inclusion of dietary counseling in individualized migraine care strategies. Table 5 and Table 6 present the suggested meal distribution (Table 5) and the recommended percentage distribution of macronutrients as sources of calories (Table 6). Figure 7 reports a summary of fundamental nutritional recommendations aimed at modulating systemic inflammation, improving metabolic resilience, and reducing the risk of chronic diseases, including emphasis on plant-based diets, restriction of pro-inflammatory foods, promotion of healthy lipid sources, and hydration optimization.

## 4. Conclusions

Migraine represents a multifactorial disorder deeply influenced by metabolic and inflammatory processes. Emerging evidence strongly supports the significant role of insulin resistance and hyperinsulinism in the pathogenesis and exacerbation of migraine symptoms. Dietary interventions, especially ketogenic and Mediterranean diets, have shown substantial potential for mitigating migraine severity and frequency by addressing underlying metabolic dysfunction, inflammation, and mitochondrial impairment. Recognizing these interconnected pathways can guide the development of personalized dietary strategies and integrated management approaches, ultimately enhancing patient outcomes and quality of life. Further research, particularly large-scale clinical trials and mechanistic studies, is essential to deepen our understanding of these relationships and refine clinical recommendations.

### Take-Home Messages

○Migraine involves complex interactions among genetic, neurological, inflammatory, and metabolic factors.○Insulin resistance and hyperinsulinism significantly contribute to migraine pathogenesis.○Metabolic dysfunction and systemic inflammation intensify migraine severity and frequency.○Targeted dietary interventions, such as ketogenic and Mediterranean diets, can effectively reduce migraine episodes.○Personalized nutritional and lifestyle approaches are crucial for optimal migraine management and improving patient quality of life.

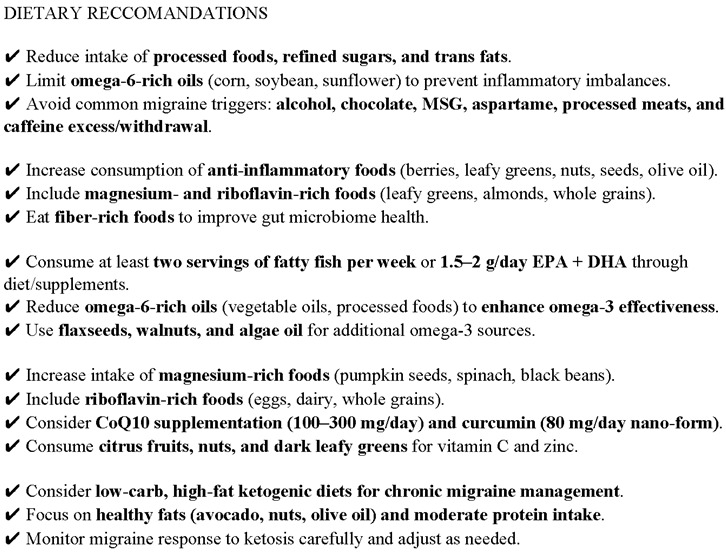



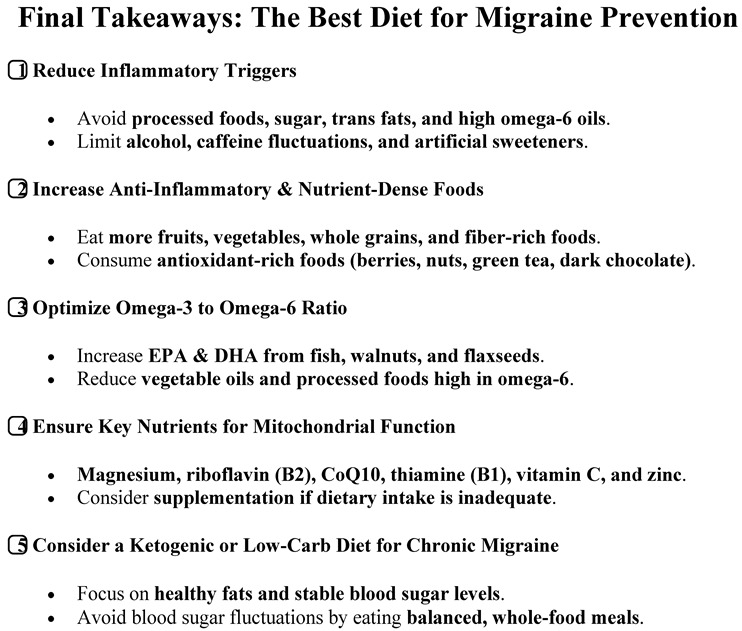



## Figures and Tables

**Figure 1 brainsci-15-00474-f001:**
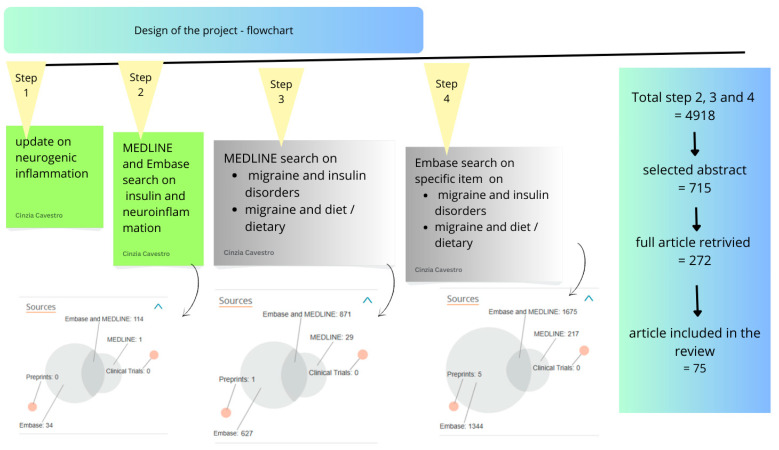
Project design—Flowchart.

**Figure 2 brainsci-15-00474-f002:**
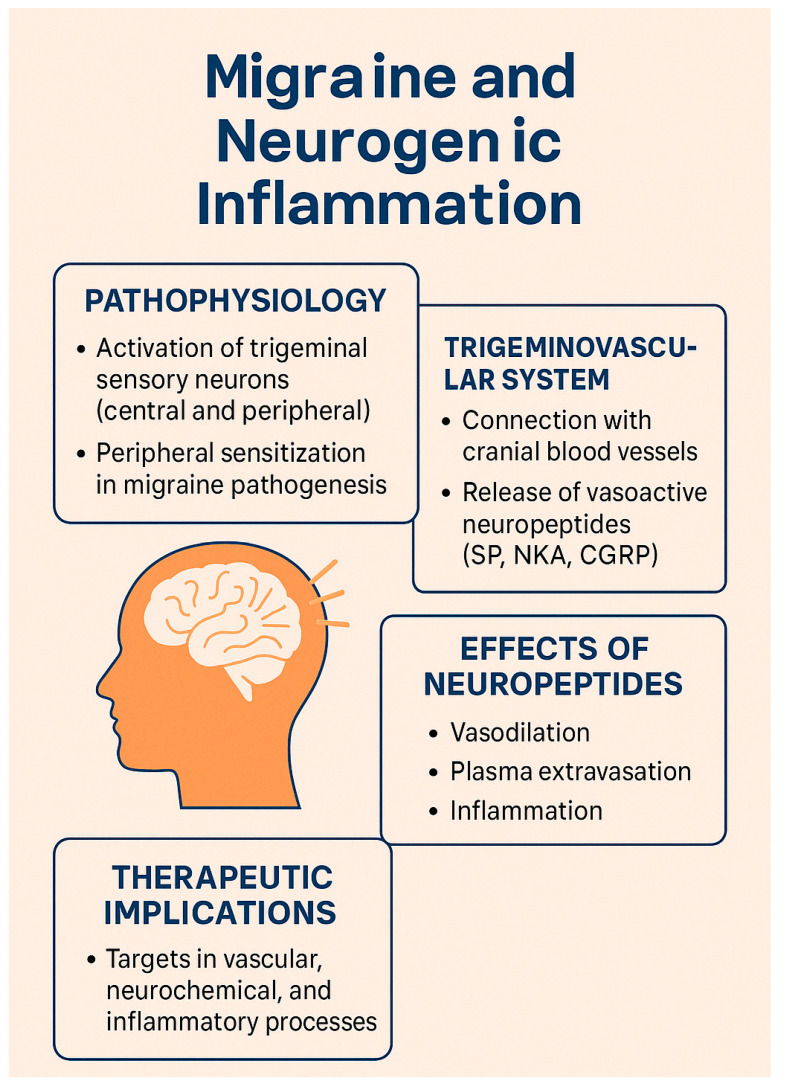
Main mechanisms involved in neurogenic inflammation in migraine.

**Figure 3 brainsci-15-00474-f003:**
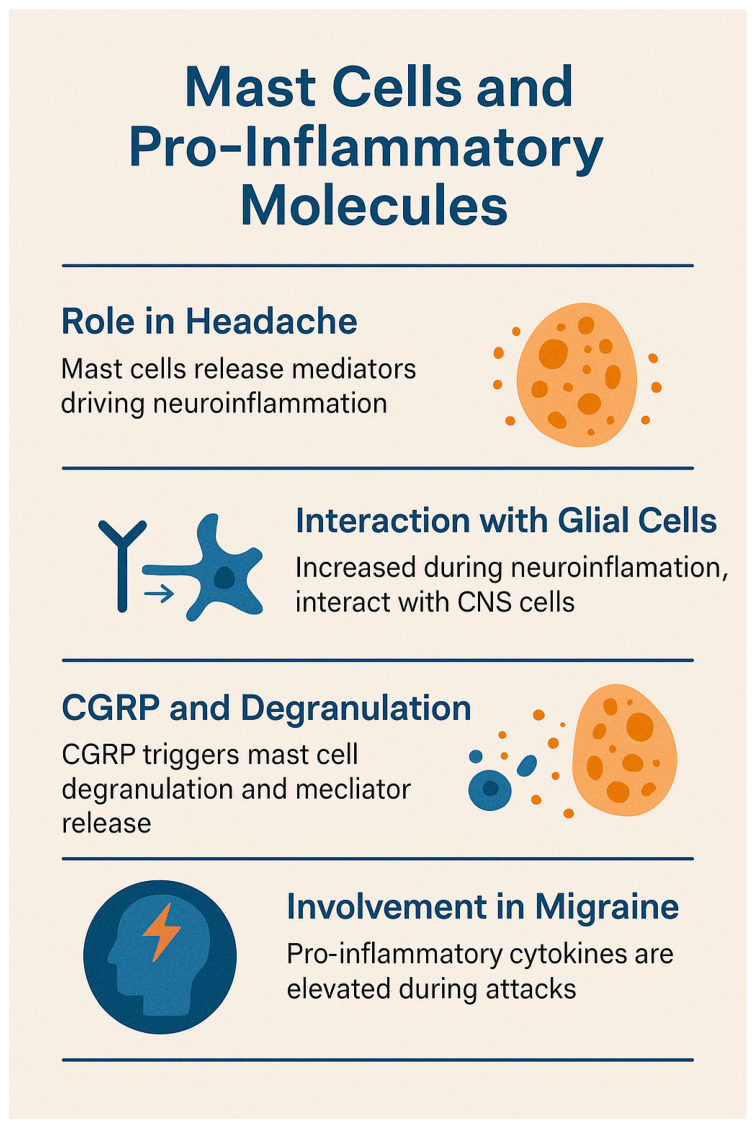
Mast cell and neurogenic inflammation in migraine.

**Figure 4 brainsci-15-00474-f004:**
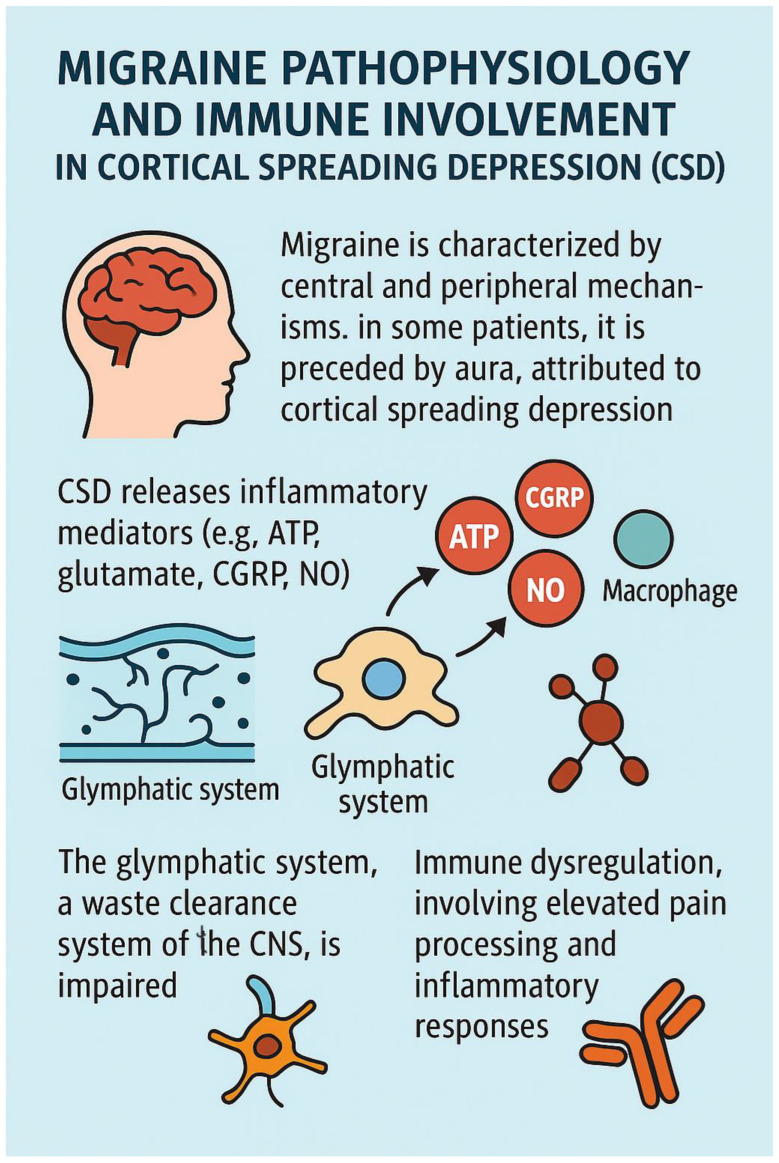
Migraine pathophisiology and immune involvement.

**Figure 5 brainsci-15-00474-f005:**
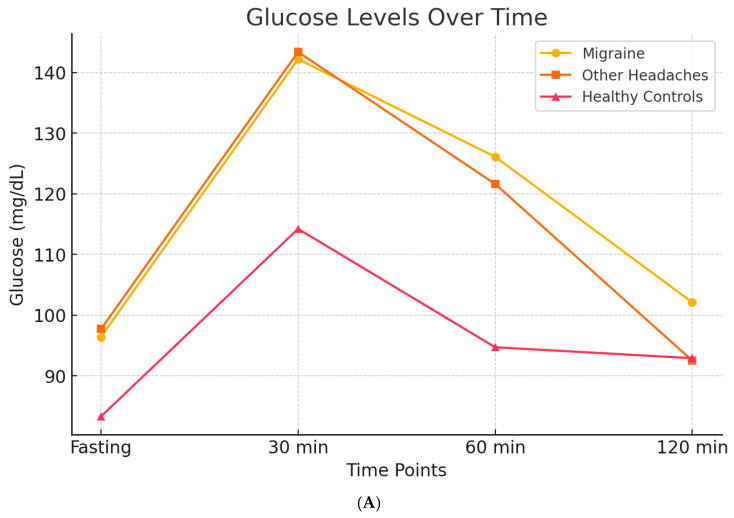
Glucose and Insulin levels over time after OGTT, in migraineurs, other headaches sufferers, and healty subjects. Subfigure (**A**) shows the trend of basal glucose values and glucose levels after a 75-g oral glucose load. Patients, both those with migraine and those with other types of headache, exhibit elevated blood glucose levels, both at baseline and after the glucose load, suggesting that headache itself may be associated with higher blood glucose levels. Subfigure (**B**) illustrates that elevated insulin levels are observed exclusively in migraine patients, when compared both to healthy controls and to patients with other types of headache. These findings suggest that hyperinsulinemia may represent a specific biological correlate of migraine.

**Figure 6 brainsci-15-00474-f006:**
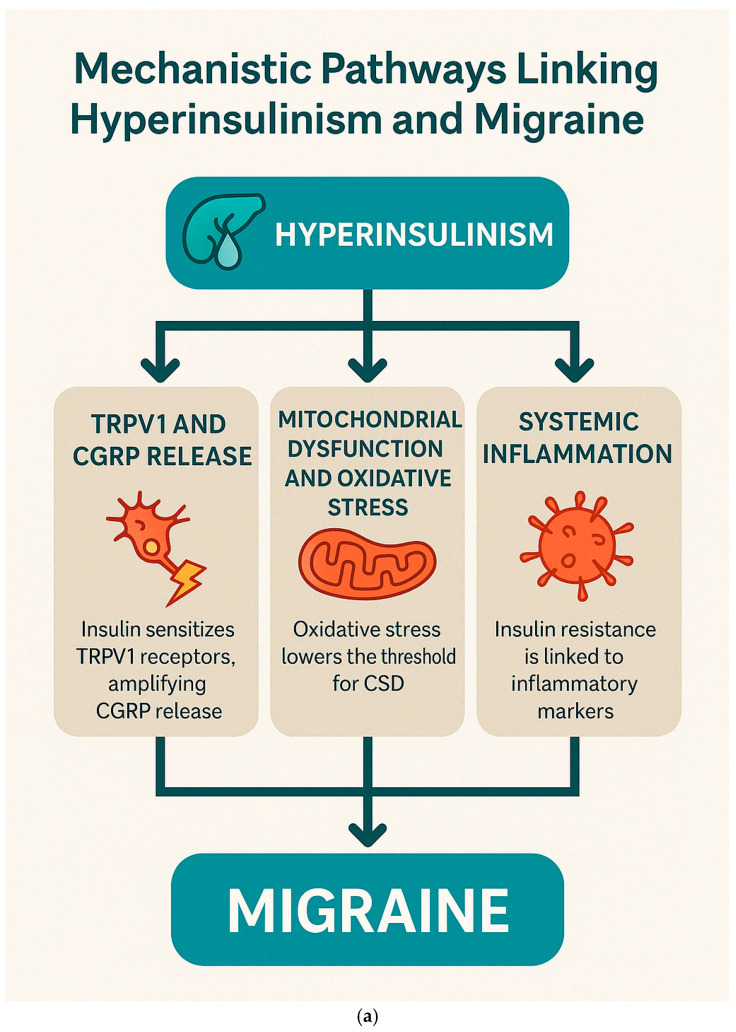
Mechanistic Pathways Linking Hyperinsulinism and Migraine. (**a**). Pathophysiological mechanisms by which hyperinsulinism contributes to migraine development via neuroinflammation, CGRP sensitization, and mitochondrial dysfunction. (**b**). “Integrative Framework Linking Hyperinsulinism to Migraine: Mechanisms, Therapeutic Targets, and Biomarkers for Personalized Management”.

**Figure 7 brainsci-15-00474-f007:**
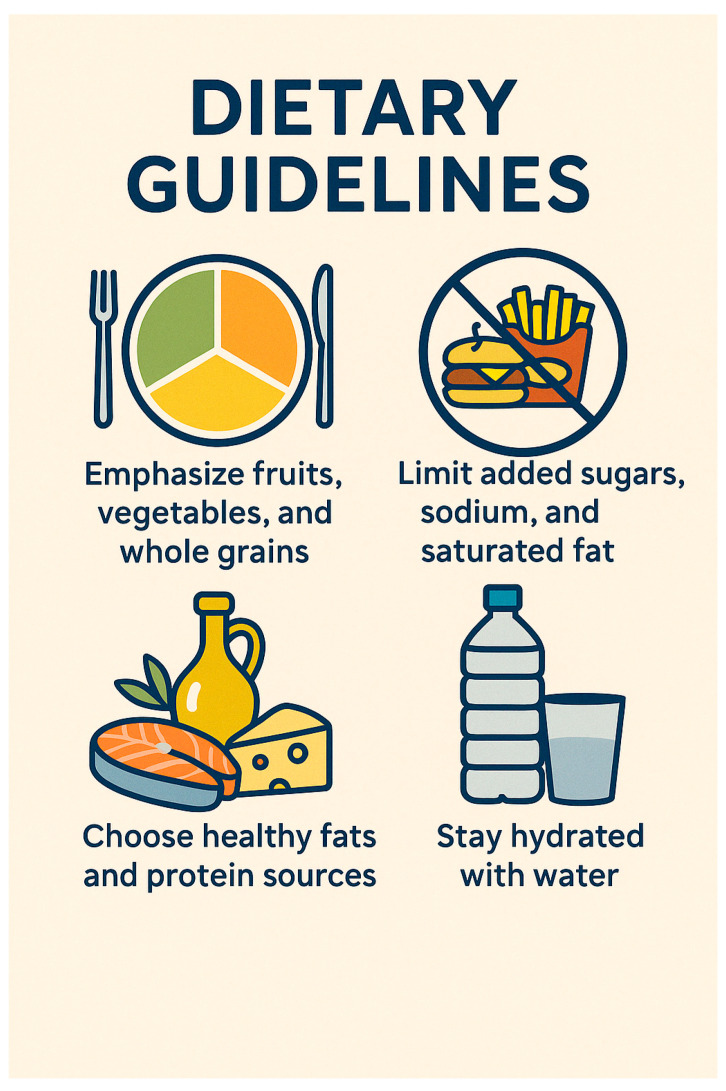
Core dietary guidelines aimed at promoting overall health and preventing chronic diseases.

**Table 1 brainsci-15-00474-t001:** Migraine mediators: mechanisms and modulation.

Substance/Peptide	Vasodilation	Plasma Extravasation	Mechanism of Action	Site of Action	Modulating Agents (Agonists/Antagonists)
CGRP	YES	NO	Acts on CGRP receptors and induces prolonged vasodilation	Cerebral and meningeal vessels	Inhibited by CGRP antagonists (gepants) and triptans
Substance P (SP)	YES	YES	Acts on endothelial NK1 receptors and induces extravasation and vasodilation	Vascular endothelium of dura mater	Inhibited by NK1 receptor antagonists
Neurokinin A (NKA)	YES	YES	Acts on endothelial receptors and induces extravasation and vasodilation	Vascular endothelium of dura mater	Inhibited by NK2 receptor antagonists
PACAP	YES	NO	Activates PAC1 receptors and induces cerebral vasodilation	Cerebral arteries (PAC1) and ganglia	Stimulated by PACAP; inhibited by PAC1 antagonists (under investigation)
Histamine	YES	YES	Inflammatory mediator and increases vascular permeability and dilation	Vascular endothelium and inflamed tissues	Inhibited by antihistamines (H1 blockers)
Serotonin (5-HT)	YES	YES	Acts on 5-HT receptors and involved in inflammation and vasodilation	C-fiber receptors and trigeminal system	Stimulated by serotonin; inhibited by triptans (5-HT1B/1D agonists)
Bradykinin	YES	YES	Inflammatory mediator and increases vascular permeability	Vascular endothelium and peripheral nociceptors	Inhibited by bradykinin B2 receptor antagonists
Prostaglandins	YES	YES	Inflammatory mediators and stimulate permeability and vasodilation	Blood vessels and inflamed tissues	Inhibited by NSAIDs (cyclooxygenase inhibitors)
Leukotrienes	YES	YES	Like prostaglandins, involved in inflammation and extravasation	Blood vessels and inflamed tissues	Inhibited by leukotriene receptor antagonists
VIP	YES	NO	Acts on VPAC receptors and serves as cerebral vasodilator without inducing migraine	Cerebral vessels (VPAC receptors)	Stimulated by VIP; not affected by standard antimigraine agents
NPY	NO	NO	Vasoconstrictor and enhances noradrenaline effect	Sympathetic system and cerebral vessels	No specific inhibitor used clinically
Ketorolac	NO	YES	Inhibits cyclooxygenase, blocks extravasation, and does not inhibit CGRP	Dura mater and brainstem	NSAIDs (e.g., ketorolac) inhibit extravasation
Aspirin	NO	YES	Inhibits cyclooxygenase and blocks neurogenic extravasation	Dura mater and brainstem	NSAIDs (e.g., aspirin) inhibit extravasation
Valproate	NO	YES	Acts on GABA A receptors and inhibits dural extravasation and transmission	Dura mater and trigeminal ganglia	Inhibits neurogenic inflammation; no upstream stimulator
Opioids	NO	YES	Acts on μ-opioid receptors, inhibiting neurogenic plasma extravasation and vasodilation	Central and peripheral µ-opioid receptors	Inhibit µ-opioid receptors (e.g., morphine)
Sumatriptan	NO	YES	5-HT1D agonist and blocks extravasation and mast cell degranulation	5-HT1D receptors on C fibers and trigeminal ganglion	5-HT1D agonist (e.g., sumatriptan) inhibits extravasation
PHM-29	YES	NO	Potent vasodilator peptide in cerebral vessels	Cerebral vessels	No clinical modulators known
Acetylcholine	YES	NO	Neurotransmitter and induces vasodilation in cerebral vessels	Cerebral vessels	No clinical modulators known
Endothelin	NO	YES	Vasoconstrictive system involved in extravasation and lacks therapeutic efficacy	Cerebral vessels and dura mater	Inhibited by endothelin receptor antagonists (ineffective clinically)
Octreotide	NO	YES	Somatostatin analog that inhibits extravasation and neuronal activation	Brainstem and somatostatin receptors	Somatostatin analog (e.g., octreotide)
Somatostatin	NO	YES	Inhibits neurogenic inflammation and neuropeptide release	Central and peripheral nervous system	Somatostatin analogs; inhibitory G-protein coupled mechanisms

**Table 2 brainsci-15-00474-t002:** Overview of insulin resistance and neurogenic inflammation.

Category	Key Mechanisms/Concepts	Implications
Insulin Resistance (IR)	Impaired PI3K/AKT signaling Chronic inflammation (TNF-α, IL-6, CRP) Oxidative stress, mitochondrial dysfunction	Hyperglycemia, dyslipidemia Risk of T2DM and metabolic syndrome
β-Cell Dysfunction	ER stress → unfolded protein response (UPR) Cytokine inhibition (IL-1β, TNF-α via NF-κB) Mitochondrial impairment, ↓ GSIS	Reduced insulin synthesis/secretion Progression of diabetes
Adipose Tissue and Adipokines	CTRP3: ↑ insulin sensitivity (via AMPK) WISP1: ↓ insulin signaling Macrophage-driven inflammation	Worsening of IR and systemic inflammation
Neuroinflammation in IR	Brain insulin resistance affects microglia Cytokine-induced IR (IL-6, TNF-α) ↓ Amyloid-β clearance	Alzheimer’s disease risk cognitive decline, depression, and schizophrenia
CGRP and Neurogenic Inflammation	CGRP promotes vasodilation, immune cell activation, and mast cell degranulation	Target in migraine; diabetic complications (neuropathy, retinopathy)
Brain Insulin Signaling	InsR on neurons, astrocytes, microglia, and endothelium Modulates GABA, NMDA, and AMPA Affects dopamine (VTA), cognition, and BBB transport	Mood, motivation, memory, and reward regulation
Therapeutic Strategies	Natural agents: curcumin and pomegranate peel (↓ NF-κB and ↑ Nrf2) CR and exercise: AMPK/SIRT1, ↑ mitochondrial function Intranasal insulin	Improve systemic and brain insulin sensitivity; ↓ neuroinflammation; ↑ cognitive function

Legend: ↑ = increase; ↓ = decrease.

**Table 3 brainsci-15-00474-t003:** Comparison of weight loss strategies and their metabolic effects.

Intervention	BMI Reduction	Effect on Migraine	Metabolic Impact
Bariatric Surgery	−10.2 kg/m^2^	Moderate to large improvement	Marked reduction in systemic inflammation, insulin resistance, and leptin levels
Behavioral Weight Loss (diet + exercise)	−4.1 kg/m^2^	Moderate improvement	Improved energy balance and reduced inflammatory cytokines
Ketogenic Diet	−3.0 kg/m^2^	Strongest migraine improvement	Increased ketone metabolism and mitochondrial efficiency
Low-Fat Vegan Diet	−1.3 kg/m^2^	Weakest effect	Minimal metabolic impact on neuroinflammation and mitochondrial function

**Table 4 brainsci-15-00474-t004:** Comparison of Gazerani and Evers on migraine and diet.

Aspect	Gazerani (2021) [127]	Gazerani (2023) [128]	Evers (2025) [129]
Diet as a Trigger	Identifies food triggers but emphasizes bidirectionality.	Re-evaluates traditional triggers, suggesting some may be prodromal symptoms.	Focuses on food as a modifiable factor via CGM-personalized guidance through DTx.
Migraine as a Metabolic Disorder	Explores mitochondrial dysfunction and energy metabolism.	Supports the ketogenic diet as a metabolic intervention.	Supports the role of glucose metabolism in migraine susceptibility and treatment response.
Dietary Interventions	Focuses on weight loss, the ketogenic diet, and omega-3.	Provides strong evidence for KD and DASH diets and personalized low-glycemic diets.	Demonstrates the significant efficacy of a personalized low-glycemic diet using DTx.
Gut Microbiota	Introduces the gut–brain axis as a key factor.	Reports mixed evidence on probiotics for migraine relief.	Not directly addressed, but suggests that dietary personalization may indirectly support microbiota balance.
Technological Advancements	Highlights the need for personalized interventions.	Advocates AI-based tracking and digital health tools.	Highlights the clinical utility of digital therapeutics and CGM-based tools for individualized dietary interventions.

**Table 5 brainsci-15-00474-t005:** Suggested meal distribution based on clinical condition.

Meal Frequency	Clinical Indications	Mechanism/Metabolic Effects
3 main meals (breakfast, lunch, dinner)	General population, normal weight, good glycemic regulation	Maintains energy balance, physiologically stimulates insulin, prevents caloric excess
3 main meals + 2 snacks	Diabetes, insulin resistance, fasting-induced headache, reactive hypoglycemia	Prevents glycemic spikes, improves satiety, reduces trigeminovascular activation
5–6 small meals/day	Metabolic syndrome, insulin resistance, migraine with metabolic component	Reduces insulin stress, enhances insulin sensitivity, prevents cortical spreading depression (CSD)
2 meals/day or intermittent fasting	Obesity, patients with high metabolic flexibility, but not suitable for migraine patients	Activates autophagy, improves insulin sensitivity, but may cause glycemic drops and migraines

**Table 6 brainsci-15-00474-t006:** Recommended percentage distribution of macronutrients as sources of calories.

Macronutrient	General Recommendations (EFSA, WHO, USDA)	For Insulin Resistance/Hyperinsulinemia (ADA; Evers (2025) [129]; García-Pérez-de-Sevilla (2025) [157])	Metabolic Mechanism/Rationale
Carbohydrates	45–60% Whole grains, fruits, vegetables, and legumes	30–40% Low-GI vegetables, legumes, oats, and limited whole grains	Lowering postprandial glycemia and insulinemia and improving insulin sensitivity (Sacks et al., 2014 [170]; ADA 2023)
Fats (Total)	20–35% Olive oil, nuts, seeds, and fatty fish	35–40% Rich in MUFAs and PUFAs (e.g., olive oil, fish); limit saturated and trans fats	Improves lipid profile and enhances insulin action; omega-3 modulates TRPV1-CGRP axis [157]
Proteins	10–20% Fish, legumes, dairy, lean meats, and eggs	20–25% Lean proteins, legumes, eggs, and fish	Enhances satiety and thermogenesis, and reduces postprandial glucose excursions [129]
Fiber	25–30 g/day Fruits, vegetables, legumes, and whole grains	30–35 g/day Legumes, leafy greens, oats, and fruits with skin	Slows carbohydrate absorption, reduces glycemic load, and improves gut microbiota [130]

EFSA: Europeans Food Safethy Autorithy; WHO: World Heath Organisation; USDA: U.S. Department fo Agriculture; ADA: American Diabetes Association.

## Data Availability

No new data were created or analyzed in this study.

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
