# Peer review of "Metabolic Dysfunction and Dietary Interventions in Migraine Management: The Role of Insulin Resistance and Neuroinflammation—A Narrative and Scoping Review"

_brainsci, 2025, doi:10.3390/brainsci15050474_

Round 1
Reviewer 1 Report
Comments and Suggestions for Authors
Dear all,
Thank you for the opportunity to review this manuscript. The manuscript aligns with the goal of the Brain Sciences, and the topic offers information for researchers, professionals, and the Neuroinflammation and Pain Medicine section. The manuscript titled presents an important topic related to migraine, insulin resistance, neurogenic inflammation, and dietary interventions. I believe the information provided might need to be clarified. Consequently, some points are listed below:
Abstract:
I recommend the author represent in the abstract the whole record has been screened (Subsequent to exclusion criteria).
Introduction
Lines (55-61); (95-99); (104-106); (111-113); (118-120); (130-133) and (136-137) need references to be supported.
At the end of this section the author mentioned that the aim of this paper is to ‘provide an updated overview of the relationship between insulin and migraine, both of them linked to inflammation, as well as to evaluate the potential therapeutic mechanisms of dietary interventions for migraine management’, without giving the readers on the abstract background of dietary habits, dietary trigger (e.g., caffeine, chocolate, monosodium glutamate, nitrate, nitrite, tyramineh) and migraine. Please give us a quick intro for these points. Also, I wish if the author could propose hypotheses to explain the relationship between insulin, migraine and inflammation.
Materials and Methods
Line 150: ‘The available literature was primarily searched using the PubMed database (all references until 2025 January)’. Could you present the year you have chosen as start date to conduct your research strategy (e.g., We focused on studies available since ……..).
Lines 160-161: ‘After excluding irrelevant and duplicate studies ….’, how did you define the irrelevant studies? What about non-English studies, case reports, and editorials? This section needs to clarify more about the search strategy, details on how articles were screened and selected are needed, I suggest using PRISMA guidelines for scoping reviews.
Results
The results need to be linked to ages (children, adolescent, adults, old persons) and gender (males, females) of participants in each study.
You mentioned and discussed some limitations that are relevant to the work, please could you add a part that will lead to new scientific studies in the future.
Conclusion
Adequate
References
The reference publication date should be bold and after the journal name in italics, please use the ACS style guide to be compatible with Brain Sciences journal guidelines.
Best wishes,
Author Response
Abstract:. I recommend the author represent in the abstract the whole record has been screened (Subsequent to exclusion criteria).
R: Methods section in abstract has been revised.
Introduction. Lines (55-61); (95-99); (104-106); (111-113); (118-120); (130-133) and (136-137) need references to be supported. At the end of this section the author mentioned that the aim of this paper is to ‘provide an updated overview of the relationship between insulin and migraine, both of them linked to inflammation, as well as to evaluate the potential therapeutic mechanisms of dietary interventions for migraine management’, without giving the readers on the abstract background of dietary habits, dietary trigger (e.g., caffeine, chocolate, monosodium glutamate, nitrate, nitrite, tyramineh) and migraine. Please give us a quick intro for these points. Also, I wish if the author could propose hypotheses to explain the relationship between insulin, migraine and inflammation.
R: proper references have been added where required, just a brief sentence is present about dietary triggers because these topic is not explored in this review.
Materials and Methods. Line 150: ‘The available literature was primarily searched using the PubMed database (all references until 2025 January)’. Could you present the year you have chosen as start date to conduct your research strategy (e.g., We focused on studies available since ……..). Lines 160-161: ‘After excluding irrelevant and duplicate studies ….’, how did you define the irrelevant studies? What about non-English studies, case reports, and editorials? This section needs to clarify more about the search strategy, details on how articles were screened and selected are needed, I suggest using PRISMA guidelines for scoping reviews.
R: all the query have been resolved.
Results. The results need to be linked to ages (children, adolescent, adults, old persons) and gender (males, females) of participants in each study. You mentioned and discussed some limitations that are relevant to the work, please could you add a part that will lead to new scientific studies in the future.
R: Some tables has been added. I did not treat what population have been studied because the aim was to find and discuss what kind of link could be hypothesised to link neurogeninc inflammation, insulin disorders and migraine. Some comment have bee added in the text under different section.
Conclusion. Adequate
References. The reference publication date should be bold and after the journal name in italics, please use the ACS style guide to be compatible with Brain Sciences journal guidelines.
R: done
Best wishes,
Many thanks
Reviewer 2 Report
Comments and Suggestions for Authors
Please find attached

Author Response
Dear authors, please find my comments below. In general, I think it is a very interesting article. However, despite some minor concerns, I wonder if it is interesting to include a so comprehensive and deeply overview of your results. And my concerns are not based on the content, since I mostly agree with everything included, but based on the readability of the manuscript.
R: I was fascinating by this topic for many years. I found it very difficult during the first years. That is why I wrote a so wide paper. I agree with you, s I wrote the content more fluently and friendly.
Title: Could you include the type of study in your title?
R: I added in the title. The paper has a part on update on neurogenic inflammatione and a second part on the scoping review, so I add in the title “a narrative and scoping review”
Abstract: Nothing to state
Introduction: Could you reintegrate everything included without splitting the content in different sections? Moreover, I would reduce the content and state only what is interesting for your study. What are you speaking about? Relationship between migraine and metabolic dysfunction. Therefore, I would perform an introduction stating epidemiologic data in migraine, mechanisms of this condition, and then integrating the possible role of metabolism. Happy to see the scope at the end of the introduction, but similar: could you integrate it with the rest of the section?
R: I integrated the sub-section in introduction, reduced the content and wrote more fluently and friendly. I added some epidemiologic data as required.
Methods: Happy to see that you included the search strategy, since many literature reviews that I’m reviewing don’t include it.
- Thanks. I develop this section as required by another reviewer
Results: You should include a figure with your results. It seems that you are mostly doing a collection of your results, rather than critical synthesis. I think that the way you presented your data is not readable and it must be modified. I would not include the results of each search one by one, but a summary of the findings. I recommend you to include some tables analyzing your results. It is TOO long.
R: I added figures and tables to make the paper more readable, shorten mainly the section on neurogenic infmammation, revised all the other section, wrote more fluently. I understand the manuscript is very long, it is more similar to a book chapter, but this topic is complex and I think all the aspect need be treated to understand the problem in his deep meaning.
Conclusions: Nothing to state.
Many thanks
Round 2
Reviewer 2 Report
Comments and Suggestions for Authors
Dear author
After the revisions, I think your article is suitable for publication.